# A framework for objectively comparing competing invasion percolation models based on highly-resolved image data

Ishani Banerjee[1¤]*, Anneli Guthke[2], Cole J.C. Van De Ven[3], Kevin G. Mumford[4], Wolfgang Nowak[1]

**1** Institute for Modelling Hydraulic and Environmental Systems (IWS)/LS3, University of Stuttgart, Stuttgart, Germany, **2** Stuttgart Center for Simulation Science, Cluster of Excellence EXC 2075, University of Stuttgart, Stuttgart, Germany, **3** Department of Civil and Environmental Engineering, Carleton University, Ontario, Canada, **4** Department of Civil Engineering, Queen's University, Kingston, Canada

¤ Current Address: Chair of Hydrogeology, Technical University of Munich, Munich, Germany
* ishani.banerjee@tum.de

## Abstract

Predicting gas migration in water-saturated porous media (a multiphase flow problem) is essential for applications such as carbon dioxide sequestration, hydrogen storage, and groundwater remediation. This process is challenging because gas migration can occur either as continuous or discontinuous flow, depending on the flow rate and the properties of the porous medium. Numerous variations of Invasion-Percolation (IP) models can simulate multiphase flow across different scales (from pore-scale to macroscopic), but the literature suggests that IP models are best suited for the discontinuous gas flow regime; other regimes have not been systematically evaluated. Here, we present a quantitative framework for comparing and ranking macroscopic IP models against high-resolution experimental data from transitional and continuous gas flow regimes. The analysis employs a diffused Jaccard coefficient to measure the similarity between model outputs and time-series two-dimensional images from gas injection experiments in water-saturated sand. To represent pore-scale heterogeneity, we run each model version on several random realizations of the initial invasion threshold field. The Jaccard coefficient was averaged over all realizations per model version to evaluate performance and calibrate model parameters. Depending on the application domain, some macroscopic IP model versions are suitable for these previously unexplored flow regimes. We also find that heterogeneity in the initial invasion threshold field strongly influences model performance. The observed applicability of IP models in these regimes can substantially reduce computational costs compared to continuum-based models for applications such as carbon dioxide storage and groundwater protection, although their ultimate use depends on the specific research question being addressed. The proposed comparison framework is not limited to gas–water systems in porous media but generalizes to any modelling situation accompanied by spatially and temporally resolved data.

**Data availability statement:** The experimental data used in this study is available in the Borealis Dataverse, titled "Replication Data for: Comparison of four competing invasion percolation models for gas flow in porous media" with doi: https://doi.org/10.5683/SP3/A7ITKL The modelling data and codes used for this study are available in the DaRUS dataverse for Stochastic Simulation and Safety Research for Hydrosystems (LS3), titled "Replication Data for: A framework for objectively comparing competing invasion percolation models based on highly-resolved image data," with doi: https://doi.org/10.18419/DARUS-3592, DaRUS, V1.

**Funding:** German Research Foundation (DFG) for financial support of this project within the Research Training Group GRK1829 "Integrated Hydrosystem Modelling" and the Cluster of Excellence EXC 2075 "Data-integrated Simulation Science (SimTech)" at the University of Stuttgart under Germany's Excellence Strategy - EXC 2075 - 39074001. The funders had no role in study design, data collection and analysis, decision to publish, or preparation of the manuscript.

**Competing interests:** The authors have declared that no competing interests exist.

## 1. Introduction

An increased understanding of gas flow processes in the subsurface is relevant for engineering applications such as subsurface carbon and hydrogen storage, groundwater contamination from leaky gas wells, and subsurface contamination (VOCs and NAPLs) monitoring and remediation. Gas flow in water-saturated porous media is a specific case of multiphase flow. The gas phase flowing through a water-saturated porous medium can be miscible or immiscible with the water phase. We explore the immiscible flow of gas in this study.

Patterns created by the immiscible flow of gas in water-saturated porous media result from an interplay between capillary forces, viscous forces, and gravitational forces [1–4]. [5] investigated the interplay between capillary forces and viscous forces for the immiscible flow of fluids in a porous medium with varying viscosity ratios. They identified three immiscible flow regimes: stable displacement (when a more viscous fluid displaces a less viscous fluid), viscous fingering (when a less viscous fluid displaces a more viscous fluid), and capillary fingering (in the absence of viscous forces). Their experiments and simulations involved multiphase flow in a horizontal setup, and the fluids used in their study did not have a considerable density contrast.

In the specific case of *gas* flow in water-saturated porous media, there is a substantial contrast in density between gas and water; thus, the influence of gravitational forces cannot be ignored. It has been observed that the interface between the fluids can be either stabilised or destabilised in the presence of gravitational forces [1,4,6–9]. For example, when a low-density fluid displaces a high-density fluid from above or when a high-density fluid displaces a low-density fluid from below in a vertical setup, buoyant forces stabilise the interface. In the other scenarios, destabilisation of the interface occurs, generating fingers (*Gravity fingering*, [10]).

When gas is injected from below into water-saturated sand, depending on the interplay between gravitational, capillary, and viscous forces, gas-water interfaces exhibit gravity fingering combined with one or more of [5] 's flow regimes. In the same porous medium, this combination depends primarily on gas injection rates. At low gas injection rates, the viscous effects are less relevant. Therefore, the flow is controlled by a combination of capillary forces (capillary fingering regime) and gravitational forces. Upon increasing the injection rates, the control shifts to a combination of viscous forces (viscous fingering regime) and gravitational forces. These gas flow regimes are classified as **continuous**, **transitional**, and **discontinuous**, depending on the grain size of the porous media and the rate of gas flow [11]. In **continuous flow** regime, the gas phase flows as a continuous phase, and in the case of **discontinuous flow** regime, gas flows as discrete gas bubbles, or clusters [6,11,12,13]. The **Transitional flow** of gas has characteristics from both the continuous and discontinuous regime. As a result of the balance of forces, the gas-flow regime tends to be discontinuous at low gas-flow rates and in coarser porous media, moving towards the continuous regime as the flow rate increases or for finer porous media [11].

Gas flow in water-saturated porous media has been investigated using gas-injection experiments in water-saturated artificial (glass beads) as well as natural (sand) porous media [e.g., 11,12,14–17, to name a few]. Besides laboratory

experiments, numerical models are often used to understand multiphase flow in porous media. These models can be essential tools to encode and test hypotheses about the multiphase flow mechanisms at work and to make useful predictions for many real-world engineering applications. Both continuum and (stochastic) discrete growth models can be used. Continuum models are fully physics-based (relying on partial differential equations) with disadvantages like being slow and computationally expensive. Discrete growth models are simplified abstractions of real systems and are fast and computationally inexpensive but have comparatively stronger underlying assumptions.

Gas flow in saturated porous media is susceptible to perturbations at the pore scale. *Continuum models* require an extremely fine mesh for the numerical discretisation to appropriately capture such local perturbations [18,19]. This further slows down the continuum-model simulations and increases their computational cost [20,18]. Modified continuum models formulated at the pore scale—using the Navier–Stokes equations to represent bubbly gas transport—have also been used to simulate discontinuous gas flow; however, this approach also has high computational cost, thus limiting their practical applicability at larger scales [21]. Both laboratory experiments and numerical model formulations of a real-world system are not free from uncertainties. While laboratory experiments can have uncertainty associated with experimental control, measurements, or data processing techniques, numerical models can suffer from conceptual and parameter uncertainty, affecting their prediction quality. Stochastic analysis of these real-world systems helps address these uncertainties appropriately. However, due to their computational cost and complexity, continuum models are not suitable candidates for such stochastic analysis.

With the rise of machine learning, data-driven methods have gained popularity as surrogates across various scientific domains, and the field of porous media is no exception, where they support uncertainty quantification by enabling more efficient approximations of complex systems [22,23]. In particular, they are becoming an attractive choice for modelling multiphase flow through porous media because of their lower computational costs in contrast to high-fidelity continuum models [24]. The applications of these methods span both large- and pore-scale problems. For instance, in large-scale carbon capture and storage problems, data-driven methods are employed to develop surrogate models for prediction and uncertainty quantification [25,26], numerically simulate $CO_2$ storage and flow behaviour [27,28,29,30], and quantify uncertainty using data assimilation and forecasting [31]. At the pore scale, these methods are used in various applications, including extracting velocity and pressure fields from flow visualisations [32] and performing direct flow simulations to assess porous media properties [33–36].

However, these methods remain approximations of full-fidelity models and typically require substantial training data. Simple process-based approaches, such as discrete growth models that incorporate the physical behaviour of the system while remaining computationally inexpensive, like data-driven methods, offer a compelling alternative. Out of many discrete growth models in the multiphase literature, e.g., Diffusion limited aggregation (DLA) [37,38], Invasion Percolation (IP) [39], anti-DLA [40] models, we are specifically interested in IP models.

Invasion Percolation (IP) models are (stochastic) discrete growth models often used for simulating displacement of immiscible fluids through porous media in the capillary fingering regime [5]. The term Invasion Percolation was first coined by [39] for a pore-scale model, which incorporated phase accessibility rules to standard Percolation models of [41] to assure connectivity within a phase.

Many IP model versions with variations in the underlying rules have been developed to match the behaviour of specific fluids in specific porous media under specific conditions [e.g., [1,7,20,42–47], to name a few]. However, all of them have the following typical conceptual and numerical implementation:

1. At first, a pore network of blocks/nodes is generated with a given connectivity by assigning each pore an invasion threshold selected from some distribution. This network can be 2D (2-dimensional) or 3D (3-dimensional).

2. Initially, all the blocks are occupied by the defending fluid. Then, the invading fluid is injected at some point in the network. For example, in our study, *water* is the *defending* fluid, and *gas* is the *invading* fluid.

3. Pores with connection to the invaded pore are evaluated for their invasion thresholds, and, based on some criterion (mostly minimum invasion threshold), one of the connected blocks is then invaded.

IP models also need to incorporate buoyancy effects to simulate gas invasion in water-saturated porous media. Several studies have therefore used IP models with gravitational/ buoyant force effects to model gas-water flow systems or fluid systems with significant density-difference in porous media [e.g., 7,8,48,49,50,45,51,46,47, to name a few]. Further, to accurately simulate gas flow from the discontinuous regime (slow gas flow rate), a rule allowing re-invasion of water into gas-filled blocks is added to the IP models [52]. This re-invasion can cause fragmentation or mobilisation of the gas clusters.

The pore-scale IP models described above must be upscaled to use them for large engineering applications like subsurface contaminant remediation, oil extraction, geologic gas storage, etc.; i.e., any scale larger than the pore-scale. Studies [e.g., 44,53,47] abstracted processes from the pore-scale IP model to then use them at the larger scales of their problems. The Near-Pore Macro-Modified Invasion Percolation (NP-MMIP) model of [20] is one such macroscopic IP model used to simulate carbon dioxide injection in a water-saturated macro-heterogeneous porous media. In the work of [54], an upscaled rule for pore-scale re-invasion of water was added to NP-MMIP to simulate gas flow in the discontinuous regime. In these macroscopic IP models, the model blocks represent a network of pores instead of single pores.

Traditional IP models, at any scale, do not incorporate viscous effects and have not been tested before in gas flow regimes other than discontinuous flow (slow-injection of gas): the transitional and continuous gas flow regimes. Experimental data from gas injection in homogeneous water-saturated sand shows that, with increasing gas injection rate, viscous forces dominate the injection zone, making the gas flow radial around the injection point [4,17]. However, once the gas propagates further away from the injection point, gravitational effects overcome the viscous effects [55]. Hence, the upward movement of gas is observed as multiple fingers (referred to as gravity fingering in [10]). Thus, at higher gas injection rates, ignoring viscous effects near the gas injection point as in traditional IP models is not a valid assumption.

The addition of several rules to IP models makes them potential candidates for transitional or continuous flow regimes. For example, [20] used an invasion of more than one block per step for their NP-MMIP model, adding more gas volume per invasion step. Further, [1] developed a generalised growth model for dense non-aqueous phase liquid (DNAPL) migration at the macroscopic scale by including invasion rules to capture viscous effects. The rule for stochastic selection in the Stochastic Selection and Invasion (SSI) model of [1] was adapted to use in simulating gas migration in water-saturated homogeneous sand [43]. Gas-injection experiments at high injection rates reveal the formation of multiple, wider gas fingers that often overlap, a phenomenon attributed to viscous effects [4,6]. Although IP models do not inherently account for viscous effects, incorporating the aforementioned rules enables the capture of the observed behaviour arising from these effects.

In general, numerical models must be compared to experimental data sets to test, calibrate and validate their underlying hypotheses, leading to their refined formulations. Although traditional macroscopic IP models are designed for use in regimes of low gas flow rate, our goal is to test their performance in the transitional and continuous flow regimes, from which direction for further model refinement can be derived. Thus, we use four models in this study:

1. NP-MMIP model of [20] without viscous modifications.

2. Macro-IP model involving the rule for re-invasion of water [43,54].

3. A combination of Macro-IP model with the rule of more than one invasion block per step (including the original viscous modification as in [20]).

4. A combination of Macro-IP model and modified stochastic selection rule of SSI model of [1] adapted from [43].

These IP models at a macroscopic scale have been compared to experiments individually and each at a certain flow regime. A systematic inter-comparison across diverse experimental datasets spanning different gas flow conditions

(across all three regimes of gas flow: continuous, transitional and discontinuous) is lacking, limiting our understanding of each model's applicability and robustness. A broader evaluation is essential to analyse the strengths and weaknesses of the models across various flow conditions, providing valuable insights for their refinement.

Thus, in this work, we test four different macroscopic IP model versions with data from nine gas-injection experiments in homogeneous water-saturated sand. These experiments belong to the transitional and continuous gas flow regimes [4], controlled by varying the injection rate. Thus, we assess the model performance under gas-flow conditions other than the discontinuous or slow-gas flow regime.

In our previous work [56], we developed and tested a quantitative method of comparison between IP-type models and laboratory gas-injection data from the discontinuous flow regime. In [56], we demonstrated our method using a single macroscopic IP model based on [43]. Now, we extend this methodology to systematically test, compare, and rank the four macroscopic IP model versions for gas flow in transitional and continuous regimes. By applying a consistent evaluation framework across multiple datasets, we aim to identify the strengths and weaknesses of the models in different gas flow conditions, guiding their further development and application in subsurface flow problems. Our key research questions are:

1. Can any of these models be used for simulating gas flow in the continuous or transitional flow regimes?

2. If yes, which ones are more suitable?

3. What can we learn from the comparison of more or less successful model strategies and their remaining weaknesses to derive recommendations for future modelling efforts?

Furthermore, the model comparison framework developed in this study is not limited to the four models tested here; it is a versatile tool that can also be applied to other models. For example, in multiphase flow problems in porous media, this framework can be applied to systems such as oil recovery and carbon dioxide sequestration (with pore network modelling, micro-CT, or capillary pressure experiments) [57]. It can also be extended to evaluate data-driven models in fields such as medical tomography [58] and digital rock physics [59]. Our work demonstrates the utility and robustness of this framework, showcasing its potential for broader application in the analysis and improvement of numerical models that predict highly resolved observable data.

We organise our model comparison study as follows. At first, we introduce the experiments and describe the formulation of the four macroscopic IP model versions used in this study in Sect 2. Then, in Sect 3, we detail the method or tool of comparison we use for evaluating and ranking the models against the experimental data. Also, we discuss the overall implementation of the method for the inter-comparison of models in Sect 3. We report the results from this implementation and provide insights about the model performance and its parameters in Sect 4. Finally, we summarise our conclusions and recommendations for future work in Sect 5.

## 2. Experiments and models

In this section, we describe the experimental data sets (Sect 2.1) and the four macroscopic IP model versions (Sects 2.2–2.3) used for our model comparison study. All four model versions are at the same scale and share some similarities. Fig 1 shows the conceptual building of the 4 model versions used in this study.

To facilitate the understanding of the models, first, we describe the model version (we call it **Model 1**) based on the NP-MMIP of [20] (Sect 2.2). Model 1 does not include the modifications for viscous effects from the NP-MMIP model of [20]. Then, in Sect 2.3, we introduce **Model 2**, which has additional rules of re-invasion of water at the macroscopic scale, same as in [43,54] (see Fig 1). **Model 3** (Sect 2.4) is a combination of Model 2 and a rule for producing thicker fingers from the viscous modification of NP-MMIP model of [20] (see Fig 1). Finally, **Model 4** in Sect 2.5, which is built by combining Model 2 and a modified rule for stochastic invasion from [1] (see Fig 1). Model 4 is based on [43]. All the model versions used here generate binary images (gas-presence/gas-absence) as output.

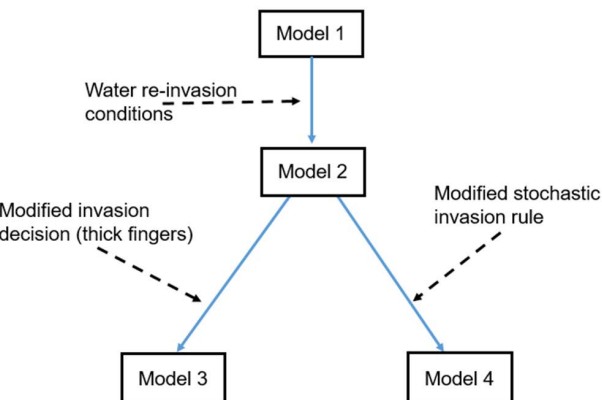

**Fig 1. Flowchart illustrating the building process of the competing model versions of this study.**

### 2.1 Experiments

For this study, we use nine gas-injection experiments from [4], which were conducted in triplicate at 10 ml/min (10-A, 10-B, 10-C), 100 ml/min (100-A, 100-B, 100-C) and 250 ml/min (250-A, 250-B, 250-C). The gas flow patterns of the different regimes are distinguished using the ratio of Bond number, $Bo$ (ratio of gravitational force to capillary force) to Capillary number, $Ca$ (ratio of viscous force to capillary force) [4]. The triplicate experiments at 10 ml/min (10-A, 10-B, 10-C) belong to the transitional flow regime, with $Bo/Ca = -1.61 \times 10^2$ [4]. The triplicate at 100 ml/min (100-A, 100-B, 100-C) with $Bo/Ca = -1.61 \times 10^1$ and at 250 ml/min (250-A, 250-B, 250-C) with $Bo/Ca = -6.45 \times 10^0$ belong to the continuous flow regime, with increasing influence of viscous forces [4]. The experimental setup and data processing details are found in [4]. We present a summary of the data relevant to understanding our study.

Gas (air) is injected in water-saturated homogeneous sand (grain size $0.713 \pm 0.023$ mm), filled into a quasi-2D acrylic cell of dimensions 250 mm $\times$ 250mm $\times$ 10 mm. A continuous wet-packing procedure was used to ensure that the resulting sand distribution was homogeneous and free of trapped gas. Air was then injected into the saturated sand packs at the defined rates of 10, 100 and 250 ml/min using a syringe pump. To ensure that no grain rearrangement occurred during injection, a confining lid was placed at the top of the system. The gas movement and resulting gas presence within the sand pack were measured using the light transmission method [60,61]. In this method, the back of the cell is lit, and digital images or videos of gas injection are recorded. In the experiments used in this study, imaging was performed using a high-resolution camera (Canon EOS 6D equipped with a Canon EF 17–40 mm lens). The videos were captured at a resolution of 1920 × 1080 pixels and a frame rate of 29.97 frames s$^1$ for the duration of each experiment. The recordings were subsequently processed to extract still-frame images, which were cropped to include only the face of the cell and then converted to grayscale intensity fields. Individual pixel intensity values of these raw images are averaged over a block size of $1 \times 1$ mm, and the intensity values of the block are used to calculate the optical density (OD) [62] values. A median filter is applied to a 3 × 3 pixel (0.75 × 0.75 mm) neighbourhood so that the resulting OD field reflects variations over areas comparable to, but not smaller than, the diameter of an individual grain. For any $1 \times 1$ mm block, OD $> 0.02$ is considered as the presence of gas. We thus obtain a time series of binary (gas/no gas) images.

Please note that, for the experimental replicates at a particular injection rate, the sand is washed and repacked using the same procedure to obtain a homogeneous packing after each experiment. Nevertheless, with a new arrangement of all grains, each experimental outcome is unique. The final time images for the nine experiments used in this study are shown in Fig 2. Note, for experimental triplicate at an injection rate of 10 ml/min (first row of Fig 2), the gas finger of 10-B

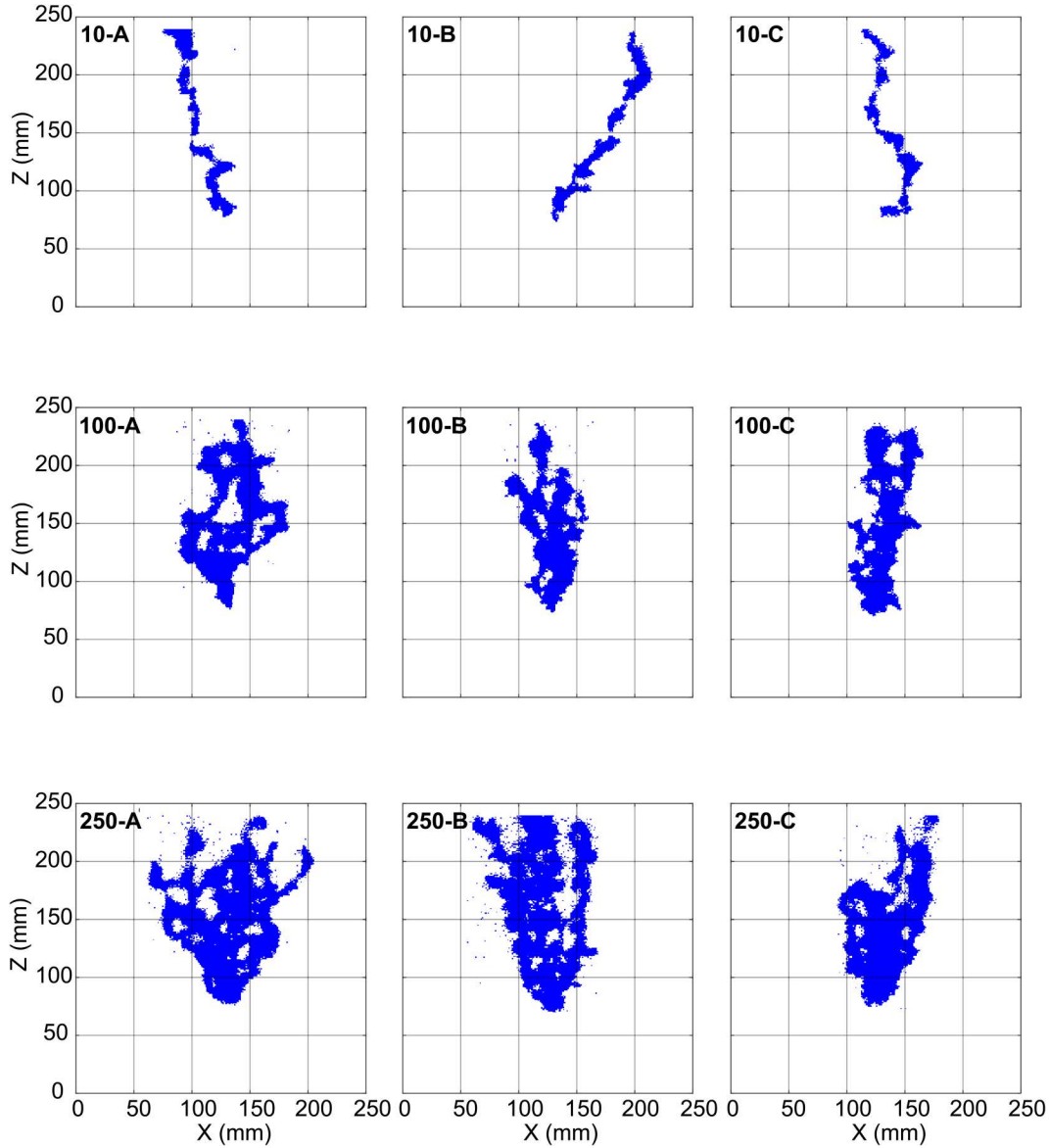

**Fig 2. Final time binary experimental images for experiments 10-A, 10-B, 10-C, 100-A, 100-B, 100-C, 250-A, 250-B, 250-C.** These gas presence/absence images are not free from pixel noise. Zones of the images where too many noisy pixels aggregate have been cleaned prior to use in this study.

moves towards the side of the domain, instead of being centrally aligned like in 10-A and 10-C. Also, for experiment 100-A (second row of Fig 2), the multiple gas fingers are quite spread out, but those in 100-C merge to produce thicker fingers along the way (second row of Fig 2). These differences in the images support the uniqueness of each experimental outcome owing to the repacking of the sand.

We use experimental datasets that involve one type of porous medium; however, this does not influence the validity or structure of the proposed comparison framework. Variations in porous media can affect individual model performance, but not the overall applicability of the model comparison framework.

## 2.2. Model 1

Our Model 1 is based on the NP-MMIP model of [20], briefly introduced in Sect 1. We adopt a 2D grid description of the porous medium in accordance with the experimental data. In this model, the gas is placed at the injection block (position of the gas injection needle in the experiment), and the invasion thresholds ($T_e$) [cm of $H_2O$] of the neighbouring blocks are calculated:

$$T_e = P_e + P_w,$$ (1)

where $P_e$ is the local entry pressure of the block [cm of $H_2O$], and $P_w$ is the pressure of the water phase [cm of $H_2O$]. $P_e$ is the specific value of capillary pressure ($P_c$) required by gas to percolate a water-occupied block. $P_w$ incorporates the buoyant effects and is calculated assuming hydrostatic conditions:

$$P_w = \rho_w g z.$$ (2)

Here, $\rho_w$ is the density of water [kg/m³], $g$ is the acceleration due to gravity [m/s²], and $z$ is the height [m] from the top of the acrylic glass cell. At each model step, the neighbouring block with the minimum invasion threshold ($T_e$) is invaded by gas.

   The $P_e$ field of a porous medium depends on the pore-scale arrangement of the solid and its interaction with the fluids. A precise measurement of the $P_e$ field at the scale of our experiments (block size of 1 mm x 1 mm) is practically impossible. Therefore, it is typical to use random $P_e$ fields, i.e., a randomly generated value per block. Since $P_e$ is a point on the capillary pressure ($P_c$)–saturation ($S$) curve, we randomly sample the $P_e$ values that we assign individually to all model blocks, using the Brooks-Corey $P_c - S$ relationship [63] for our material of interest (homogeneous sand of 0.7 mm average grain size):

$$S_e = \left(\frac{P_c}{P_d}\right)^{-\lambda}.$$ (3)

Here, $S_e$ is the effective wetting phase saturation [63], $P_c$ is capillary pressure [cm of $H_2O$], $P_d$ is the macroscopic displacement pressure [cm of $H_2O$], and $\lambda$ is the pore-size distribution index. The value of $\lambda$ varies typically between 1–4 and can be up to 7 for very uniform sands. We sample the $P_e$ values from the inverse of the cumulative distribution function of $P_c$ (using Equation 3):

$$P_e = P_d \mathcal{U}^{-\frac{1}{\lambda}}.$$ (4)

Here, $\mathcal{U}$ is a random number from the standard uniform distribution on the interval [0,1]. This sampling method is called the Inverse Transform sampling method, which has been used in the works of [20,43,56]. The $P_e$ values thus assigned to the blocks are not spatially correlated, but this extension could easily be achieved via geostatistical simulation.

## 2.3 Model 2

Our Model 2 has the same setup and follows the same rules for invasion of gas as specified for Model 1 (Sect 2.2). This means it follows Equations 1–4 and also obeys the rule of invading the neighbouring block with the minimum $T_e$. Furthermore, it has a rule for re-invasion of water into gas-occupied blocks to simulate the fragmentation and mobilization events observed for discontinuous gas flow [43,56,54]. This rule is an upscaled version of the re-invasion rule of the pore-scale model of [52].

   In [52], the re-invasion of water into the gas-filled pores is realized by a withdrawal pressure threshold. At the scale of our model, the threshold for re-invasion, also known as the terminal threshold ($T_t$) [cm of $H_2O$], is calculated as the summation of the terminal pressure ($P_t$) [cm of $H_2O$] and the hydrostatic pressure ($P_w$).

$$T_t = P_t + P_w. \tag{5}$$

$P_t$ is calculated using the $P_e-$ to $-P_t$ ratio ($\alpha$) obtained from the characteristic drainage and imbibition curves for the porous medium of interest, which takes capillary-pressure hysteresis into account [12,64].

$$P_t = \alpha P_e \tag{6}$$

Water re-invades a gas-occupied block if:

$$T_{t,g} > T_{e,w}, \tag{7}$$

where $g$ and $w$ stand for gas- and water-occupied blocks, respectively [43]. In the model, this rule is implemented by comparing the maximum of the $T_{t,g}$ values of the gas cluster with the invasion threshold value of the most gas invasion favourable neighbouring water-occupied grid block (minimum $T_e$ value). When water re-invades a gas-occupied block, the model assumes that it completely expels gas from that block. If the re-invasion of water occurs in blocks on the periphery of the gas cluster, mobilization occurs. If the re-invasion causes a disconnection in the gas cluster, fragmentation occurs. A gas cluster is allowed to grow (based on the rules of Model 1) only when connected to the gas cluster containing the injection point. Thus, only rearrangement of blocks is possible for gas clusters disconnected from the injection point.

## 2.4 Model 3

Our Model 3 includes an invasion rule of [20] into our Model 2 implementation. In this regard, our model formulation follows the rules specified by the Equations 1–7. The difference is that multiple neighbouring blocks ($nb$) are invaded instead of one block per step. Consequently, instead of invading only the block with the lowest invasion threshold, the $nb$ lowest-threshold candidates are selected. This reduces the controlling influence of $T_e$ and thereby reflects a shift from capillary-dominated behaviour towards a regime in which viscous effects become more prominent, as is characteristic of gas flow at higher injection rates. The number of blocks to invade is chosen by observing the gas fingers from the experimental data.

Please note that, in our implementation, the number of blocks invaded is chosen dynamically until the number of blocks specified at the beginning of the simulation is available for invasion. For example, in a model run specified to invade $nb = 10$ blocks per step, initially, when the number of available neighbours is $< 10$, all the available ones are invaded. Ten neighbouring blocks are invaded only when the gas cluster around the injection point is big enough to have $\geq 10$ neighbouring blocks. After the invasion of multiple blocks, fragmentation and mobilization are carried out in a similar manner as described in Model 2. This means that the simulation of the fragmentation and mobilization event in Model 3 does not involve gas invasion of multiple water-occupied neighbouring blocks.

## 2.5 Model 4

Model 4 is implemented following the formulations specified by Equations 1–7. Model 2 selects the neighbouring block with a minimum invasion threshold ($T_e$) for invasion. In contrast, in Model 4, the neighbouring block is chosen using a modified rule for stochastic selection from the Stochastic Selection and Invasion (SSI) model of [1]. This rule allows gas to invade not strictly only the block with the minimum invasion threshold ($T_e$) but also less easy-to-invade blocks based on a partially randomized choice. The distinction between Model 3 and Model 4 is that Model 3 weakens the strict control of $T_e$ by advancing several blocks at each step, whereas Model 4 does so by selecting a single block stochastically. In both cases, relaxing the strict ordering of thresholds shifts the invasion pattern away from a purely capillary-dominated process and towards behaviour in which viscous effects play a stronger role, as is expected at higher gas-injection rates.

In the modified rule for stochastic selection:

1. The list of $T_e$ values of the neighbouring blocks ($n$) of the gas cluster are arranged in an ascending order $T_{e,asc}$ and the cumulative sum $T_{e,cum}$ is evaluated:

$$T_{e,cum}[i] = \sum_{j=1}^{j=i} T_{e,asc}[j]; \, i = 1, 2, 3, \ldots, n.$$

(8)

2. Then the first block (value of $i$) where the rule specified by Equation 9 is found true is invaded by the gas:

$$T_{e,cum}[i] > \mathcal{R}^c \sum_{j=1}^{j=n} T_e[j].$$

(9)

Here, $\mathcal{R}$ is a uniformly distributed random number between [0,1] and $c$ is the cell selection weighting factor [1]. Please note that although $\mathcal{R}$ and $\mathcal{U}$ from Equation 4 are from the same distribution, their seed numbers and generator types are different. Hence, we use different symbols here.

In the stochastic selection rule, $c$ controls the strength of randomness, and its value lies in the range of $(0, \infty)$. When $c \to \infty$, the value of $\mathcal{R}^c \to 0$ for almost all values of $\mathcal{R}$. In this case, the first block on the list of $T_{e,asc}$ (block with the lowest $T_e$ value) will be invaded deterministically by gas. The resulting lightning-bolt-like gas finger is the same as the gas finger generated by Model 2. In fact, for $c \to \infty$, Model 4 becomes identical to Model 2. However, the lower the $c$ value, the higher the RHS of Equation 9, which ensures that the higher $T_e[j]$ are picked more often; this generates gas fingers that are not moving strictly upward, but have a wider spatial distribution. Please note that the re-invasion of water events that result in fragmentation or mobilization of gas clusters are carried out exactly as in Model 2, i.e., without any stochastic modification.

Table 1 shows the model parameter values used in this study.

The conceptual difference in the model versions is illustrated using a schematic in Fig 3. Fig 3b displays a gas invasion event in Model 1, which gives rise to a lightning-bolt-like gas finger. The fragmentation of the gas cluster owing to water re-invasion, as per Model 2, is shown in Fig 3c. Fig 3d shows the gas invasion of three blocks (three most favoured blocks according to $T_e$ values) in the injection cluster following a fragmentation event, according to Model 3. Fig 3e displays the invasion of a randomly chosen neighbouring block (not the most favourable block according to the $T_e$ values) following a fragmentation event according to Model 4.

We will show outputs generated by the Models $1-4$ with the best fit to experimental images from 10-A, 100-$A$ and 250-$A$ in Sect 4.

## 3. Method of comparison

We begin with a summarized description of our comparison method (Sect 3.1), the details of which are in [56]. Then, we list the blur-radii chosen for the Diffused Jaccard coefficient in this study in Sect 3.2. After that, we enumerate the steps of our model comparison study using the (Diffused) Jaccard Coefficient in Sect 3.3.

### 3.1. Experiment-model comparison by (Diffused) Jaccard Coefficient

In [56], we developed a method to compare IP-type models to image-based data. We used the method to compare a macroscopic IP model (Model 2 of this study) with a gas-injection experimental data set from the discontinuous regime.

**Table 1. Model parameters used in this study.**

| Parameters [Units] | Symbols | Values |
|---|---|---|
| **Common for models 1–4** | | |
| Density of water [kg/m³] | $\rho_w$ | 1000 |
| Acceleration due to gravity [m/s²] | $g$ | 9.82 |
| Average $P_t - P_e$ ratio [-] | $a$ | 0.6 [12] |
| Displacement pressure [cm of $H_2O$] | $P_d$ | 8.66 [65] |
| Pore-Size distribution index [-] | $\lambda$ | 5.57 [65] |
| Model domain size [mm²] | $X - Z$ | $250 \times 250$ |
| Block discretization [mm²] | $x - z$ | $1 \times 1$ |
| **Model 3 specific** | | |
| Number of blocks to invade | $nb$ | {1,2,...10, 15,20} for experiments at 10 ml/min |
| | | {1,2,...20, 25, 30, 35, 40, 50} for experiments at 100 ml/min and 250 ml/min |
| **Model 4 specific** | | |
| Cell selection weighting factor | $c$ | {5, 10, 15, 200, 500} |

Comparing IP-type models to laboratory or field data is challenging because they do not involve a time description. We overcome this challenge by implementing a volume-based time matching, where the volume of gas at each time step of the experiment ($V_{exp}$) is evaluated:

$$V_{exp}(t) = \sum_{t=t_{exp}}^{t=t_{end}} Q_{exp} \times t; t = t_{exp}, 2 \cdot t_{exp}, 3 \cdot t_{exp}, ...t_{end},$$

(10)

and volume of gas per model loop counter ($V_{model}$) is evaluated as:

$$V_{model}(n_c) = \sum_{n_c=1}^{n_c=n_{top}} n_{blocks} \times \phi \times S_g \times V_{block}; n_c = 1, 2, 3, ...n_{top}.$$

(11)

Here, $Q_{exp}$ is the gas-injection rate of the experiment [volume/time], $t_{exp}$ is the time step in between the capture of two successive images in the experiment, $t_{end}$ is the time when the experiment ends, $n_{blocks}$ is the number of blocks invaded per loop counter $n_c$ of the model, $n_{top}$ is the model loop counter when the gas reaches the top of the domain, $V_{block}$ is the volume of each discretized block of the model, $\phi$ is the porosity, and $S_g$ is the gas saturation value assigned to the entire gas cluster based on the values observed in the experiments [56]. We search the nearest neighbour in the $V_{exp}$ vector for all the time-wise elements in the $V_{model}$ vector. Then, we assign the experimental time to the corresponding nearest-neighbour model loop counter.

After the volume-based time matching of the model output and the experimental data, we use the (Diffused) Jaccard coefficient to assess the fit quality between the model and the experimental data (images). As per the set theory, for two sets A and B, the Jaccard coefficient ($J$) is defined as:

$$J(A, B) = \frac{|A \cap B|}{|A \cup B|}.$$

(12)

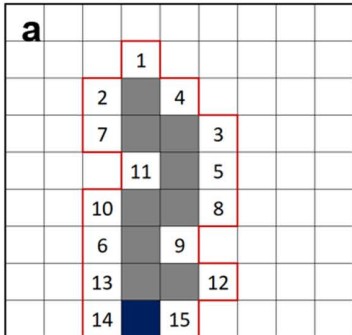

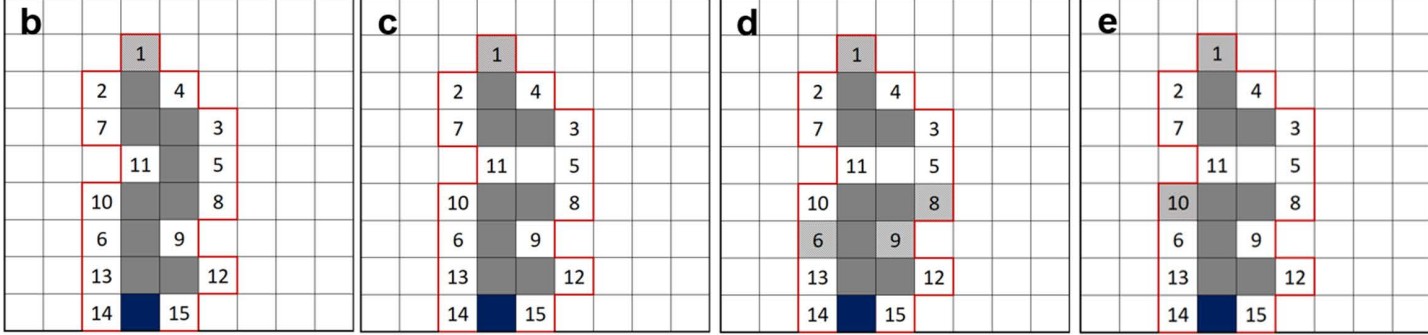

**Fig 3. Illustration of the conceptual difference between the four model versions: a** is an initial state of gas occupation in the domain, and the numbers denote the increasing order of preference of gas invasion for the neighbouring blocks in the next step based only on $T_e$ **values; b displays gas filling in the next step according to Model 1; c displays fragmentation of gas cluster in the next step according to Model 2; d** displays a fragmentation event followed by an invasion event involving three invasion blocks. **(nb = 3) according to Model 3; e displays a fragmentation event followed by an invasion event according to Model 4.** Light grey cells are the blocks chosen by the respective model version, and the blue block is the injection site.

The Jaccard coefficient ranges between zero (implies: no similarity) and one (implies: complete similarity). For binary images (pixel values of gas present = 1 and gas absent = 0), it is calculated by counting the number of overlapping pixels (value 1) between two images and dividing it by the combined total number of gas presence (value 1) pixels in both the images, without double counting the already overlapped pixels (see [56] for details).

A pixel-by-pixel comparison as in Equation 12 could reject a perfect model due to minor offsets between experiment and model, which might be within the tolerance of some real-world applications [56]. To avoid a strict pixel-by-pixel comparison of the images, we use a Diffused Jaccard coefficient ($J_d$) instead of the Jaccard coefficient. To compute the Diffused Jaccard coefficient, we blur the time-matched images from the experiment and the model using Gaussian blurring by convoluting the images with a Gaussian kernel of specified width (standard deviation $\sigma$):

$$G(x, z) = \frac{1}{2\pi\sigma^2} e^{-\frac{x^2+z^2}{2\sigma^2}},$$

(13)

The $\sigma$ value in Equation 13 is altered to increase or decrease the blurring radius. We specify the unit of blur-radius as the kernel size relative to the original domain size of the image. The blurring leads to non-binary pixel values in the images. Therefore, we evaluate the Diffused Jaccard coefficient ($J_d$) for the sets $A = \{a_k : a \in R, k = 1, 2, ...n_p\}$ and $B = \{b_k : b \in R, k = 1, 2, ...n_p\}$ using the non-binary formulation of the Jaccard coefficient (also referred to as Ruzicka similarity coefficient [66]):

$$J_d(A, B) = \frac{\sum_k^{n_p} \min(a_k, b_k)}{\sum_k^{n_p}, \max(a_k, b_k)}$$

(14)

where $a_k$ and $b_k$ are the grey-scale values of the originally black-white (binary) images from experiments and models. For simplicity, we restrict our analysis to the final (last in time) experimental images and the corresponding model images.

### 3.2. Blur-radii for diffused Jaccard Coefficient

Further, we choose three different blur-radii for the Diffused Jaccard coefficient as a performance metric for ranking the models in this study.

1. **Low blur:** We choose this blur-radius such that images from the experiments (see, Fig 2) lose the sharpness of the pixels but do not lose their identity, i.e., the different blurred experimental images look different. This corresponds to any application where we forgive errors in individual pixel values but insist on a close match in shape (Low blur row of images in Fig 4). The chosen value of $\sigma$ for this blurring is 1.2% of the domain size, i.e., image width. The Diffused Jaccard coefficient calculated using this blur radius is denoted as *Diffused Jaccard coefficient (low)* ($J_d^{low}$) in this study.

2. **Medium blur:** We choose this blur-radius such that images from the experimental triplicate at any injection rate (each row of Fig 2) look similar, but that the images across different injection rates look different. This corresponds to applications where it is sufficient to identify diversion by flow-inhibiting structures and the overall direction of the growing finger (Medium blur row of images in Fig 4). The chosen value of $\sigma$ for this blurring is 4% of the domain size. Please note that it is not entirely attainable, e.g., when a finger, like in experiment 10-B, favours a particular direction of flow, no amount of blurring can make it look like fingers from 10-A or 10-C where the flow is clearly in the centre of the cell. The Diffused Jaccard coefficient calculated using this blur radius is denoted as *Diffused Jaccard coefficient (med)* ($J_d^{med}$) in this study.

3. **High blur:** We choose this blur-radius such that images from all the experiments (Fig 2) lose the individual details in finger structure and start looking similar. This corresponds to any application where one is interested only in the macroscopic direction of the gas finger and in no further details (High blur row of images in Fig 4). The chosen value of $\sigma$ for this blurring is 8% of the domain size. Please note again that the images from all experiments cannot look the same with any meaningful blur radius. The higher flow rates have multiple fingers and more gas in the system and can thus handle more blurring than the lower injection rate experiments that generate a single finger. The Diffused Jaccard coefficient calculated using this blur radius is denoted as *Diffused Jaccard coefficient (high)* ($J_d^{high}$) in this study.

In Fig 4, we show the resulting images of the experiments 10-A, 100-A, and 250-A, with and without the blurring.

### 3.3. Steps of model comparison study

We present an overview of the model-comparison setup in Fig 5.

We have four competing model versions as described in Sects 2.2–2.5. In step ②, we run the models over several (500) invasion threshold ($T_e$) realizations for all model versions (including the sub-versions discussed below) to appropriately account for the uncertainty involved with the invasion threshold ($T_e$) fields.

Prior to this, step ① requires some parameter specifications. We run Model 3 (Sect 2.4) for varying numbers of blocks to invade ($nb$) at each step, creating many sub-versions of this model to test the best-fitting value. At injection rates of 100 ml/min and 250 ml/min, we expect a higher number of blocks to perform well because there is a high volume of gas injected into the system. We set the range of $nb$ by visual inspection. For the experiments at injection rate of 10 ml/min, $nb$ takes the values {2,3,4,...10, 15, 20}. We assign values of {2,3,4,...20, 25, 30, 35, 40, 50} to $nb$ for the experiments

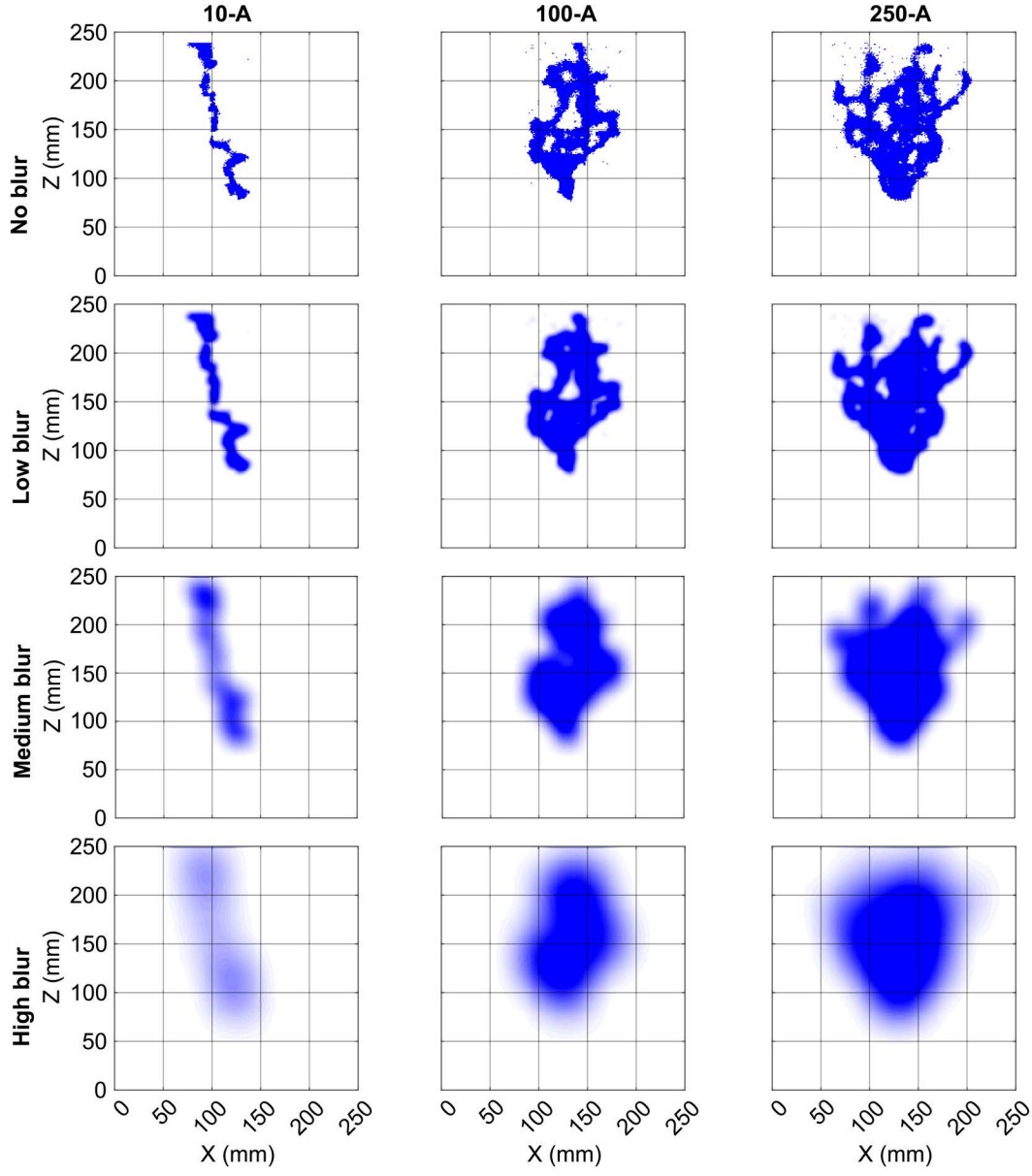

**Fig 4. Final experimental image for experiments 10-A, 100-A and 250-A.** Row 2-4 contains the blurred version of the images of Row 1 for the three different blur-radii.

at injection rates of 100 ml/min and 250 ml/min. Please note that larger $nb$ values (>50 blocks per step) would lead to inflated circular shapes instead of multiple gas fingers, and hence $nb=50$ was set as the upper limit.

Further, we run Model 4 (Sect 2.5) for some representative $c$ values: {5,10,15,200,500} creating five sub-versions of this model to test the best-fitting value. We suppose that, while the transitional flow regime (10 ml/min) would prefer higher $c$ values (200 or 500), the continuous flow regime (100 ml/min and 250 ml/min) would prefer low $c$ values, because low $c$ values allow the gas to spread more laterally instead of strictly moving upwards. Please also note here that we ran the simulations for $c<5$ values as well. However, this did not lead to systematic improvements or more insightful results, so

 

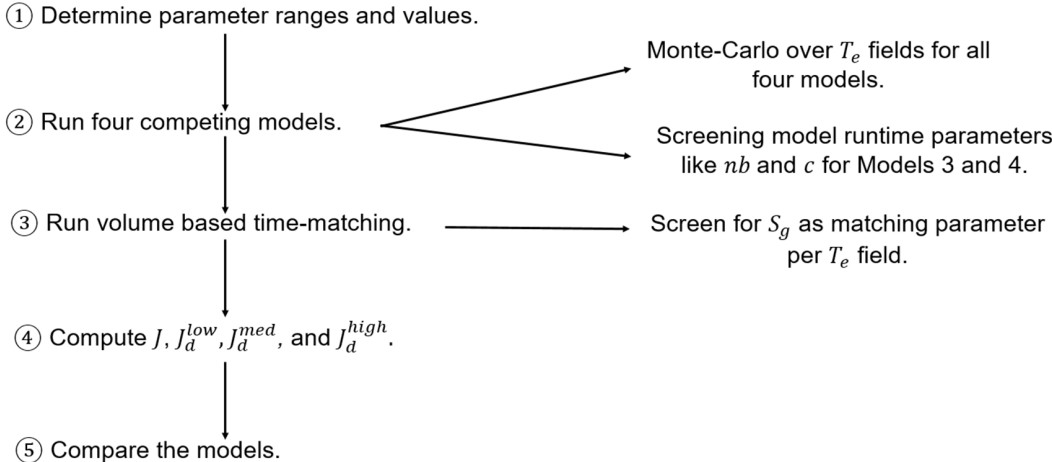

**Fig 5. Flow chart listing the steps of the model-comparison setup.**

we excluded them from further analysis due to their very long runtime. Further, this study does not aim to formally optimize the $c$ value for specific model variants with an extensive search over the feasible parameter space.

In step ③, we run the time-matching procedure for all the model versions and sub-versions mentioned above. Additionally, to calibrate gas saturation values assigned per block of the model domain within the time matching, we conduct the time-matching by varying the $S_g$ values in Equation 11 in the range of $0.02 - 0.44$ (in accordance with experimentally observed gas saturation values of [55]). In step ④, we compute the $J$, $J_d^{low}$, $J_d^{med}$, and $J_d^{high}$ values to assess the quality of fit between the experimental images and the corresponding time-matched model images. Per $T_e$ field realization, we want the model to choose its most suitable saturation value based on the maximum metric value. Also, these metrics are used to compare the performance of the competing model versions.

## 4. Results and discussion

We assess the performance of all four models (Sects 2.2–2.5) and comment on their ranking (Sect 4.1) for the different experiments (from Sect 2.1) using the Jaccard coefficient and Diffused Jaccard coefficients enumerated in Sect 3.2. In our discussion, we use the term "metric" to address the Jaccard coefficient and the three levels of the Diffused Jaccard coefficient (low, med, high) altogether. We further support our deductions from the metric-based ranking by visual evidence in Sect 4.2. In Sect 4.3, we discuss the importance of the random invasion threshold fields as model input. Also, we discuss the results from calibration of the gas-saturation parameter in the models in Sect 4.4. Finally, we summarise our findings from this model selection study in Sect 4.5

### 4.1. Overall ranking of models

We begin the discussion by commenting on the overall ranking of the competing models based on the maximum metric value out of the 500 $Te$ field runs. The table specified by Fig 6 shows that for all metric values and across most experiments, Model 1 and Model 2 rank poorly compared to Model 3 and Model 4. This is entirely expected for the experiments of the continuous flow domain (with injection rates 100 ml/min and 250 ml/min) because Model 1 and Model 2 do not include rules incorporating the gas-fingering behaviour (viscous effects, multiple fingers etc.) at these injection rates.

In the transitional flow domain (10 ml/min experiments), gas flow behaviour already shows characteristics of the continuous flow regime [4], where capillary forces do not entirely dominate over the viscous forces (Sect 1). Recall from Sects

| Triplicate Experiments | Injection rate | 10 ml/min | | | | 100ml/min | | | | 250ml/min | | | | Metric |
|---|---|---|---|---|---|---|---|---|---|---|---|---|---|---|
| | Models | 1 | 2 | 3 | 4 | 1 | 2 | 3 | 4 | 1 | 2 | 3 | 4 | |
| | A | 0.207 | 0.187 | 0.297 | 0.225 | 0.110 | 0.106 | 0.446 | 0.381 | 0.083 | 0.080 | 0.439 | 0.422 | Jaccard coefficient |
| | B | 0.135 | 0.129 | 0.168 | 0.178 | 0.142 | 0.137 | 0.494 | 0.392 | 0.090 | 0.086 | 0.535 | 0.408 | Jaccard coefficient |
| | C | 0.141 | 0.138 | 0.185 | 0.161 | 0.144 | 0.137 | 0.486 | 0.366 | 0.107 | 0.103 | 0.417 | 0.423 | Jaccard coefficient |
| | A | 0.338 | 0.325 | 0.473 | 0.372 | 0.133 | 0.130 | 0.541 | 0.453 | 0.096 | 0.092 | 0.521 | 0.496 | Diffused Jaccard coefficient (low) |
| | B | 0.234 | 0.227 | 0.308 | 0.278 | 0.201 | 0.191 | 0.612 | 0.488 | 0.109 | 0.098 | 0.644 | 0.491 | Diffused Jaccard coefficient (low) |
| | C | 0.265 | 0.260 | 0.320 | 0.271 | 0.173 | 0.169 | 0.620 | 0.474 | 0.134 | 0.130 | 0.490 | 0.491 | Diffused Jaccard coefficient (low) |
| | A | 0.493 | 0.474 | 0.713 | 0.628 | 0.164 | 0.154 | 0.670 | 0.604 | 0.112 | 0.104 | 0.605 | 0.605 | Diffused Jaccard coefficient (med) |
| | B | 0.384 | 0.364 | 0.490 | 0.471 | 0.238 | 0.218 | 0.747 | 0.615 | 0.122 | 0.110 | 0.758 | 0.623 | Diffused Jaccard coefficient (med) |
| | C | 0.463 | 0.449 | 0.539 | 0.478 | 0.203 | 0.188 | 0.784 | 0.578 | 0.148 | 0.142 | 0.572 | 0.610 | Diffused Jaccard coefficient (med) |
| | A | 0.527 | 0.501 | 0.821 | 0.700 | 0.175 | 0.158 | 0.758 | 0.753 | 0.120 | 0.107 | 0.674 | 0.700 | Diffused Jaccard coefficient (high) |
| | B | 0.458 | 0.422 | 0.639 | 0.617 | 0.244 | 0.218 | 0.827 | 0.715 | 0.122 | 0.110 | 0.842 | 0.705 | Diffused Jaccard coefficient (high) |
| | C | 0.584 | 0.551 | 0.725 | 0.709 | 0.216 | 0.194 | 0.873 | 0.663 | 0.152 | 0.145 | 0.652 | 0.633 | Diffused Jaccard coefficient (high) |

better ↔ worse

**Fig 6. Table containing the maximum metric value for each model version out of the 500 *Te* field runs and for the best gas-saturation ($S_g$) value (see Sect 4.4).** For Model 3 and Model 4, the metric corresponds to the respective best parameter value (see Table 2).


2.2 and 2.3 that Models 1 and 2 do not account for viscous effects and are completely formulated to be operated in the slow gas flow regime (discontinuous flow). Therefore, we note that the contrast in performance between Models (1,2) and (3,4) is higher for higher injection-rate experiments (the difference in metric values is higher for 100 ml/min and 250 ml/min in the table specified by Fig 6). On that account, for the entire transitional and continuous flow regime, we do not recommend the use of Model 1 and Model 2. Overall, in our study, Model 3 emerges as the best-performing model for most experiments and metrics, always (and often closely) followed by Model 4.

The blurring of the images does not change the overall ranking of the models across all investigated scales of interest. The difference in the model outputs occurs (e.g., finger width, finger direction, etc.) even on larger scales. We discuss the effect of blurring further when we discuss the models' relative performance across all 500 $T_e$ field realisations (see Sect 4.1.2).

**4.1.1. What about the Parameter Values of Models 3 and 4?.** Models 3 and 4 have additional parameter values *nb* and *c*, respectively, that have been tested on a range of values (see Sect 3.3). In Table 2, we report the parameter values corresponding to the best-performing metric values of Fig 6, i.e., again for the best-performing $T_e$ field per model.

As anticipated in Sect 3.3, at injection rates of 100 ml/min and 250 ml/min, Model 3 performs best with a higher number of blocks of invasion (see columns of 100 ml/min and 250 ml/min in Table 2). For Model 4, the best performing *c* values for injection rates of 100 ml/min and 250 ml/min are indeed the smallest on the list: $c=5$ (see columns of 100 ml/min and 250 ml/min in Table 2), as already predicted in Sect 3.3.

We observe that, for the injection rate of 10 ml/min, the best *c* values of Model 4 also correspond to the ones contributing to more inner randomness, i.e., the ones that assist in the radial spreading of the gas. This is unexpected at first sight: At an injection rate of 10 ml/min, viscous effects exist but are not predominant, i.e., we observe less radial spreading in the experiments (top row of Fig 2). We have observed similar behaviour in one of our earlier works [67], where the experimental data belonged to the discontinuous gas flow regime.

**Table 2. Table containing the values of the best respective parameter value for Models 3 and 4 for the best-performing gas-saturation ($Sg$) value (see Sect 4.4), i.e., number of blocks ($nb$) for Model 3 and $c$ values for Model 4. The evaluation is based on Jaccard coefficient ($J$), Diffused Jaccard coefficient (low) ($J_d^{low}$), Diffused Jaccard coefficient (med) ($J_d^{med}$), and Diffused Jaccard coefficient (high) ($J_d^{high}$).**

| | Injection rate | 10 ml/min | | 100 ml/min | | 250 ml/min | | |
|---|---|---|---|---|---|---|---|---|
| | Models | 3 | 4 | 3 | 4 | 3 | 4 | |
| | Parameters | $nb$ | $c$ | $nb$ | $c$ | $nb$ | $c$ | |
| Triplicate Experiments | A | 8 | 10 | 50 | 5 | 50 | 5 | $J$ |
| | B | 3 | 15 | 40 | 5 | 50 | 5 | |
| | C | 5 | 5 | 30 | 5 | 50 | 5 | |
| | A | 8 | 10 | 40 | 5 | 50 | 5 | $J_d^{low}$ |
| | B | 3 | 15 | 35 | 5 | 50 | 5 | |
| | C | 5 | 5 | 30 | 5 | 50 | 5 | |
| | A | 6 | 15 | 40 | 5 | 50 | 5 | $J_d^{med}$ |
| | B | 3 | 5 | 35 | 5 | 50 | 5 | |
| | C | 3 | 200 | 30 | 5 | 40 | 5 | |
| | A | 5 | 15 | 40 | 5 | 50 | 5 | $J_d^{high}$ |
| | B | 4 | 5 | 35 | 5 | 50 | 5 | |
| | C | 3 | 10 | 30 | 5 | 40 | 5 | |

Two opposing arguments are relevant to understand these surprisingly low $c$ values at 10 ml/min. On the one hand, the higher $c$ values (200 or 500) for a given invasion threshold are almost deterministic in their choice of the gas path. When these $c$ values meet the invasion threshold ($T_e$) field closest to the actual experiment conditions, the model can accurately produce the gas path with the highest similarity to the observed experimental gas finger. But for any threshold field with poor resemblance to the actual experimental conditions, models with these high $c$ values produce poor-fitting gas fingers. On the other hand, models with lower $c$ values are more flexible in their choice of a gas path for a given invasion threshold field ($T_e$). Combining the two arguments, these best-performing low $c$ values indicate that, in the absence of a good fit of the structure of the $T_e$ field to the experimental porous medium, the more flexible models fare well.

### 4.1.2. Relative Performance of the Models across 500 Runs.

Until now, we have discussed the model performance based on the overall maximum metric value out of the 500 runs. To analyse the relative performance of the model versions and sub-versions (with varying parameters, see Sect 3.3) across 500 runs per metric value, we inspect the percentage of ranks obtained by each of them. We present a few plots to aid our discussion in Figs. 7 and 8. Please note that these rankings are relative among the models (and model sub-versions) per individual experiment, and it thus does not indicate whether any of these models best fit the experiments used in this study.

We observe from the rank-plots of experiments 10-A, 10-B, and 250-A using the Jaccard coefficient (Fig 7, top row, and Fig 8 top), that the Models 1 and 2 rank mediocre to poor amongst all the model (sub-) versions. Further, we notice that the best model according to the overall maximum metric value (Model 3, see table specified by Fig 6) does not consistently rank well for all the 500 $T_e$ fields (This becomes visible by the presence of red colour in the bars of Model 3 sub-versions in Fig 7 and 8). This indicates that the $T_e$ field is an essential input for these models, which will be further discussed in Sect 4.3.

Also, we notice that Model 4 with larger $c$ values representing more systematic behaviour (relying primarily on the $T_e$ field) ranks the best for 10-A (e.g., see bars 4c200 or 4c500 of the top row, left plot in Fig 7), and those with $c$ values representing somewhat directionless randomness to partially overrule the $T_e$ field, rank better for 10-B (e.g., see bars 4c5 or 4c10 of the top row, right plot in Fig 7). In the experimental results of 10-B, the gas finger moves towards the right boundary of the domain, indicating the significant influence of the $T_e$ field in this experiment compared to 10-A where the

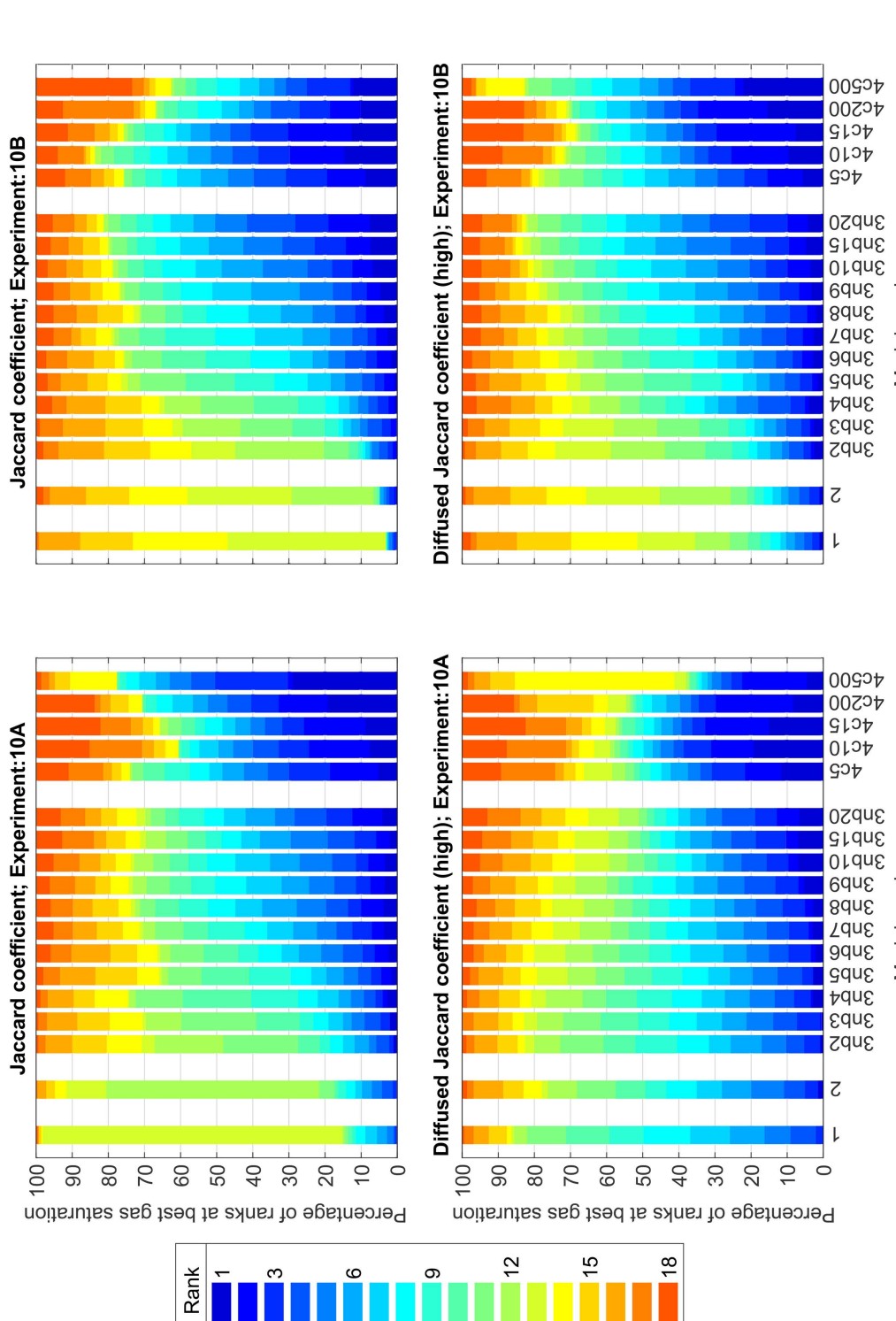

**Fig 7. Bar plot of the percentage of relative ranks obtained by each model version out of the 500 runs for the best-performing gas-saturation value for the corresponding run.** The plots are for experiment numbers 10-A and 10-B, and the corresponding metrics used for ranking are mentioned in the title of the subplots. Labels 1 and 2 correspond to Models 1 and 2 of this study. The label 3nb2, 3nb3,…. stands for Model 3 with $nb=2,3,…$ invaded blocks and the label 4c5, 4c10,…. stands for Model 4 with $c=5, 10,…$ respectively.

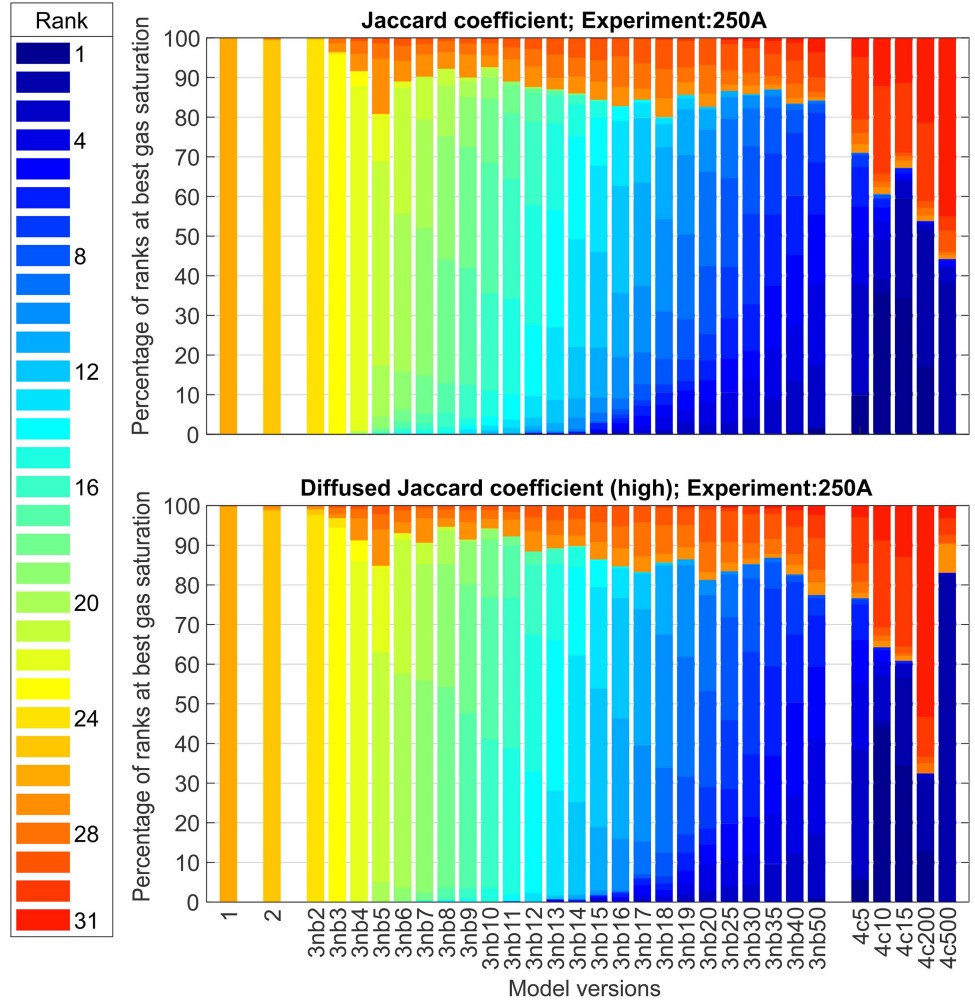

**Fig 8. Bar plot of the percentage of relative ranks obtained by each model version out of the 500 runs for the best-performing gas-saturation value for the corresponding run.** The experiment number 250-A and the corresponding metric used for ranking are mentioned in the title of the sub-plots. Labels 1 and 2 correspond to Models 1 and 2 of this study. The label 3nb2, 3nb3.... stands for Model 3 with $nb = 2,3,...$ invaded blocks and the label 4c5, 4c10,... stands for Model 4 with $c = 5$, 10,... respectively.

gas moves through the centre of the domain (see Fig 2). The probability of a random $T_e$ field leading to a good match with that of experiment 10-B is extremely low. To overcome this large uncertainty in the $T_e$ field in our models, the more flexible models (with more randomness at lower $c$ values) perform better. In conclusion, the $T_e$ field matters for all models investigated here.

For higher injection rates, Model 4 with different $c$ values ranks the best for some realisations and worst for others (e.g., the red-blue bars from the top plot in Fig 8). This confirms our earlier impression that these models have gas finger patterns resembling the experimental images only when accompanied by "good " $T_e$ fields. With $T_e$ fields far away from that of the experiment, these models perform the worst. Hence, the "very good" Model 4 is highly sensitive to the $T_e$ field input.

Blurring the images (i.e., comparisons at larger scales) makes the ranking less strict. Even weak models like 1 and 2 rank well for a higher percentage of times (see bottom row plots in Fig 7) than they do for the non-blurred image comparison, i.e., using the plain Jaccard coefficient. However, for a high injection rate, blurring cannot help these models improve

their ranking (bottom plot for Fig 8) because the models are missing surrogate processes for viscosity, which is essential in this flow regime. In this regard, the extensions proposed in Models 3 and 4 perform well.

## 4.2. Detailed discussion of the model selection results

We further support the rankings observed in Sect 4.1 with more visual evidence and provide insights into the performance of the individual model (with its best $T_e$ field).

Comparing the images (both blurred and non-blurred) of experiment 100-A and 250-A of Fig 4 to outputs from Model 1 and Model 2 (Fig 9), one can see that they are incapable of producing branched gas-finger patterns resembling those from experiments at higher injection rates. Even with a high blurring radius, Model 1 and Model 2 produce patterns very different from the experiments at 100 ml/min or 250 ml/min simply because they are incapable of having high volumes of gas in the domain.

We refer the reader to the supplementary information in this manuscript for more visual evidence. Supplementary Figs S2 Fig – S3 Fig show experimental images and their blurred versions for experiments 10-B/C, 100-B/C, and 250-B/C. S4 Fig - S5 Fig present the best-fitting model realisations based on the maximum Jaccard coefficient for the same set of experiments. Figs S6 Fig – S14 Fig display best-fitting model realisations obtained using the maximum Diffused Jaccard coefficients at low, medium and high diffusion levels for experiments 10-A/B/C, 100-A/B/C, and 250-A/B/C.

Model 3, which emerges as the best model for almost all the metrics and experiments in Sect 4.1, has more gas in the system (with many gas-occupied blocks in the domain) (Row 3 and columns 2 and 3 of Fig 9). This is why it matches the higher injection rate experimental images better than Models 1 and 2.

The experimental images for triplicate at any particular injection rate differ in structure. Even with very high blurring, experimental images from 250-A (Fig 4) and from 250-C (S2 Fig) have different patterns. This difference is not observed in the respective best-fitting outputs from Model 3 (see Fig 10 and S13 Fig). The gas finger patterns produced by Model 3 are hardly distinct from one another (see Fig 10).

Model 4, due to the inherent randomness in the invasion decision, can have many gas-occupied blocks within the domain (Row 4 and columns 2 and 3 of Fig 9), facilitating a lateral spread of gas. However, unlike Model 3, it produces distinctive patterns. For example, in Fig 10, the best-fitting Model 4 outputs to the various blurred versions of the experimental image of 250-A are not all alike. Note that although the patterns are distinct, they are not always completely similar to the experimental image.

Therefore, we again recommend that Model 1 and Model 2 should not be used for transitional or continuous gas flow regimes. Model 3 can be used for the transitional gas flow regime (with single, slightly thick fingers). At higher flow rates with many-branched fingers (continuous flow regime), Model 3 can be used at large scales (with blurring), but *with caution*: Model 3 is not capable of differentiating between different gas cluster shapes and structures. Thus, using Model 3 in the continuous regime will likely misrepresent gas volumes, pathways, and gas-water contact with associated effects on storage and mass transfer estimates. The close runner-up model (Model 4) is a suitable candidate for use in transitional and continuous flow regimes (identifying the different shapes of gas clusters), but the underlying rules need to be modified to closely match the gas flow processes involved at high injection rates, which is beyond the scope of the present work.

## 4.3. Importance of the invasion threshold fields

The analysis across 500 $T_e$ realisations in Sect 4.1.2 demonstrates that model performance is highly conditional on the invasion threshold field. Deterministic model variants perform exceptionally well only for a narrow subset of favourable $T_e$ fields, while more flexible formulations show greater robustness when the $T_e$ field poorly represents the experimental medium. This sensitivity confirms that $T_e$ is a critical input for the models.

Recall that each of the best-performing metrics in Fig 6 corresponds to a best-fitting $T_e$ field. *Are there any similarities in the structures of these otherwise random best-fitting $T_e$ fields for the different models?* We try to identify one path of least

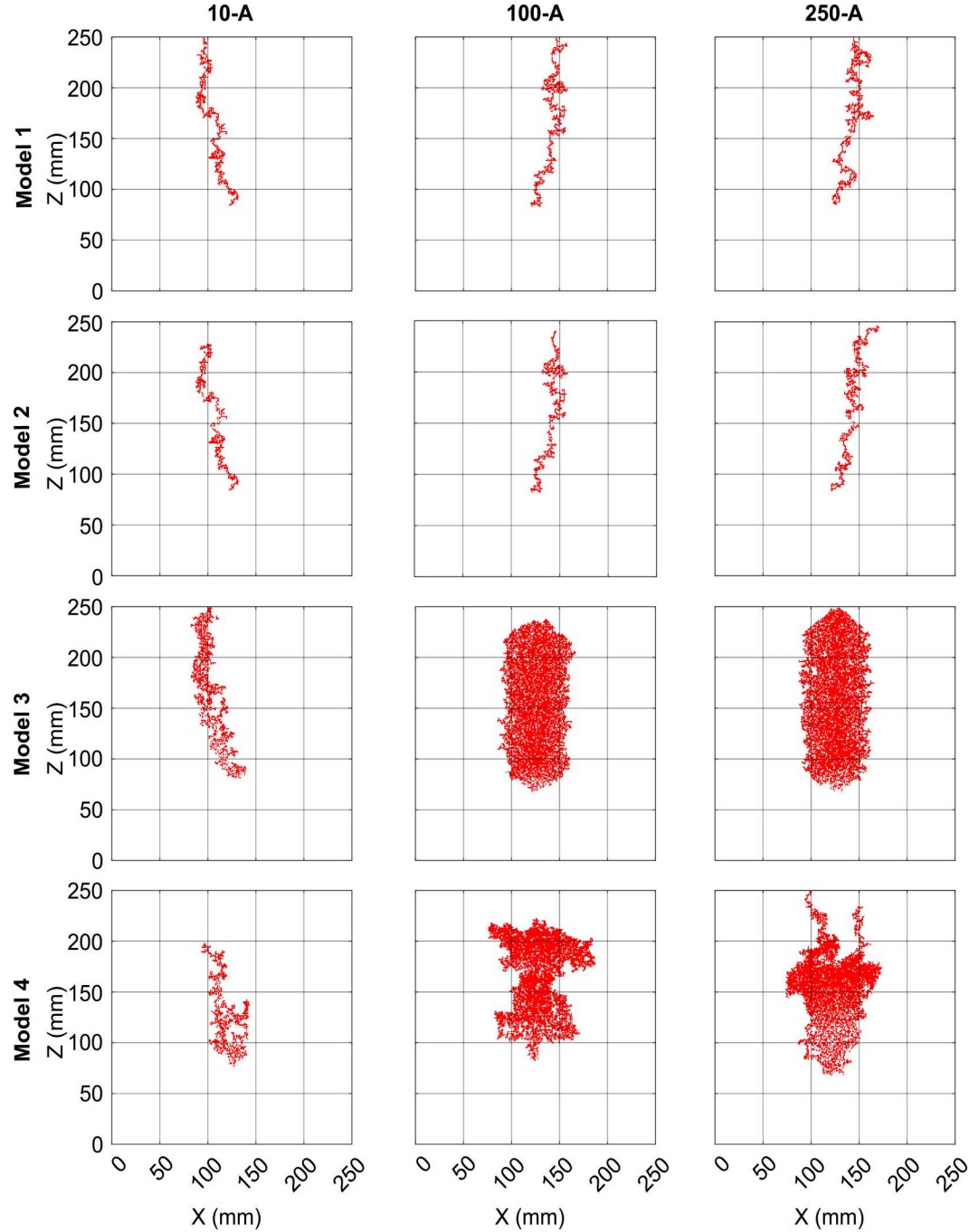

**Fig 9. Model images for the different model versions with the best fit to non-blurred experimental images (with maximum Jaccard value) from experiment no. 10-A, 100-A and 250-A.** Row 1, Row 2, Row 3 and Row 4 correspond to Model 1, Model 2, Model 3 and Model 4, respectively.

resistance through the $T_e$ fields by running Model 1 on them. This means that Model 1 runs on the best $T_e$ field for each model version evaluated using the maximum Jaccard coefficient. We choose Model 1 because, in it, all parameters except the $T_e$ field are assumed to be fixed. The overlay of the so-obtained gas fingers on the experimental image shows that

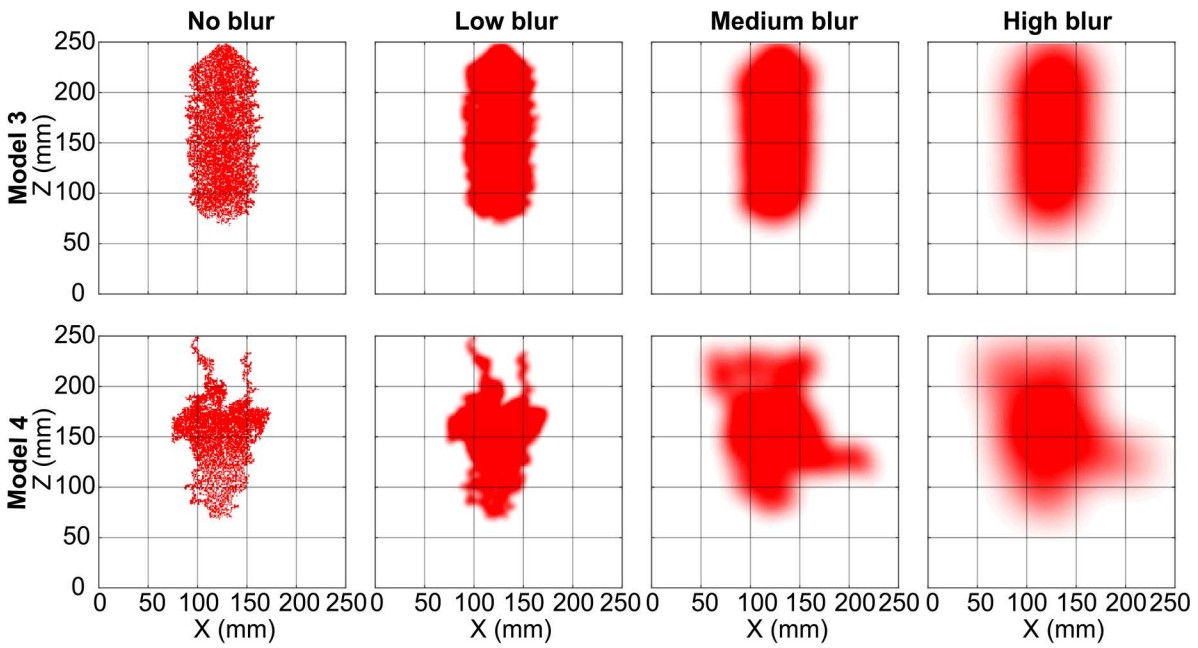

**Fig 10. Best-fit model images for Models 3 and 4 relative to non-blurred and blurred versions of experimental image 250-A.**

they partially cover the actual paths of the gas finger (Fig 11). This answers the question pertaining to the similarities in the underlying structure of the best-fitting $T_e$ fields.

Further, this observation (from Fig 11) provides strategies to handle the importance of the $T_e$ fields in spite of its uncertainty for these models. The strategy of [45] was to run their IP model over multiple realisations of their $T_e$ field to account for the uncertainty of the geological heterogeneity in their experimental setup. This seems a viable approach in this regard. Additionally, our comparison metric can be used to identify the "good performing" $T_e$ fields for each model type. One could operate a (geostatistical) Bayesian inference to estimate (or conditionally simulate) the $T_e$ fields, e.g., using Markov chain- Monte Carlo (MCMC) methods for random fields [68], a parameter Ensemble Kalman filter (EnKf) (e.g., Kalman Ensemble generator by [69]) or transformed versions [70].

### 4.4 Best-fitting gas saturation values

Recall that the results presented in the table specified by Fig 6 used the best-fitting gas saturation values ($S_g$) resulting from the time matching procedure per model and realisation (of $T_e$ field). Now, we investigate these best-fitting $S_g$ values out of our proposed range for each model per metric (Sect 3.3). Remember that our experimental data and model outputs are binary (gas-presence/gas-absence) images. The gas saturation values are an overall value provided to the entire gas cluster, i.e., all gas blocks in the binary image are replaced by the same gas saturation value. Varying the gas-saturation value varies the $V_{mod}$ in Equation 11, thus altering the corresponding time-matched image from the model outputs. Thus, the value of the metric changes when we change the gas-saturation value. In Table S1 Table, we present the best-performing gas-saturation values corresponding to the best metric values for the three experimental triplicate (table specified by Fig 6). While some of the gas-saturation values reported in Table S1 Table are comparable to those found in the experimental data, some are infeasible. For example, a value of $S_g = 0.02$ (appears multiple times in Table S1 Table) for the entire gas cluster is clearly too low.

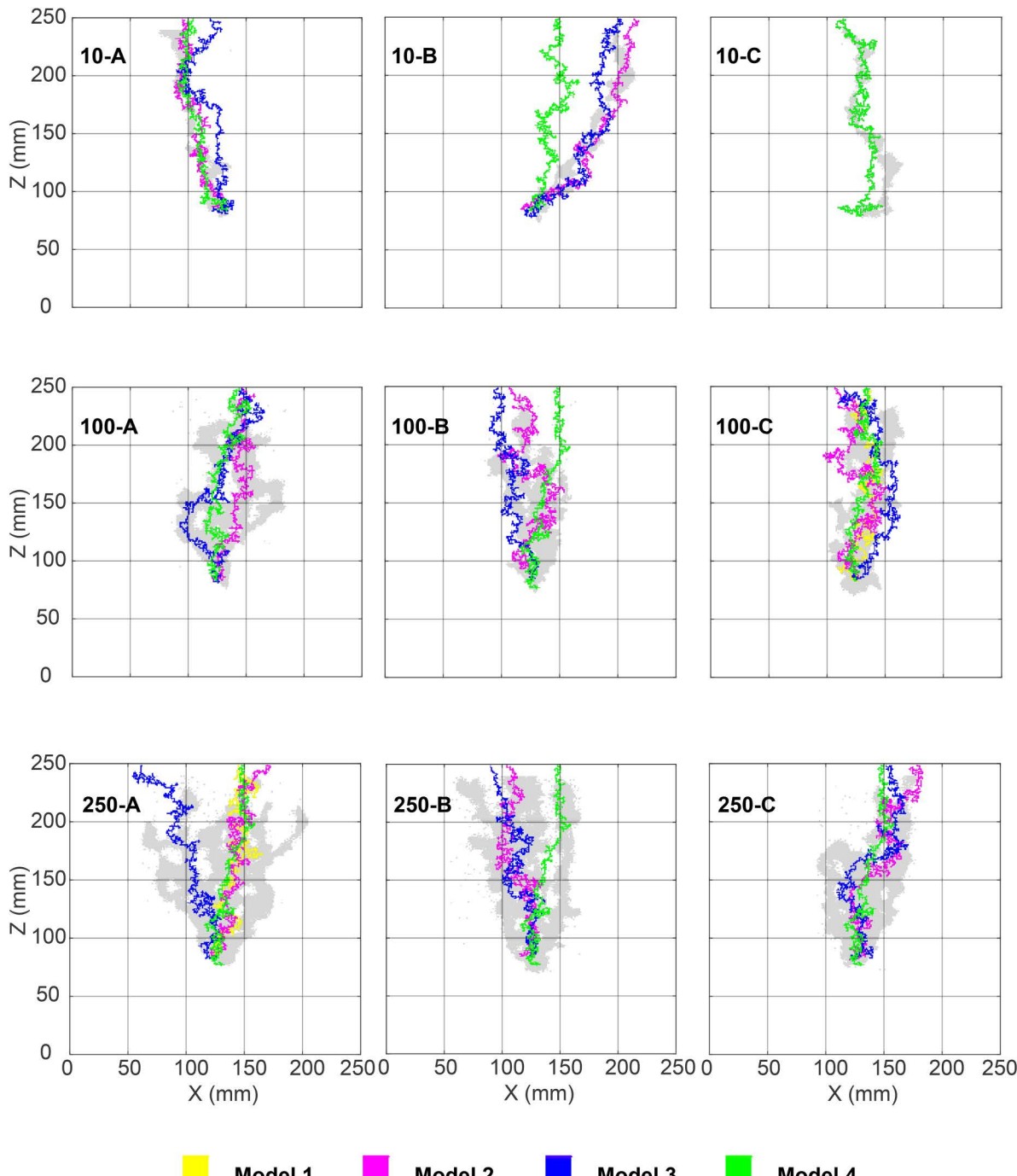

**Fig 11. Figure shows the $T_e$ field chosen for the maximum Jaccard coefficient per model version.** It is produced using Model 1, in which only $T_e$ fields vary; the other parameters are constant. Grey-coloured gas fingers represent the experimental image. Please note that each of the nine images has five different coloured fingers. The colours that are not visible in any of the sub-images are due to the overlap of pixels.

We further investigate the distribution of the gas saturation ($S_g$) values per model (sub-) version for all 500 $T_e$ field realisations. For that, we present a sample of nine scatter plots for $S_g$ (matched per $T_e$ field realisation) versus the metric (Jaccard coefficient and Diffused Jaccard coefficient (high)) for selected models (Model 1, Model 3 and Model 4) and experiments 10-A, 100-A, and 250-A in Fig 12. We pick the sub-versions of Models 3 and 4 with the best-performing parameter values, $nb$ and $c$, for the corresponding cases (see Table 2).

There is no clear optimal value of $S_g$, i.e., the values do not show a cluster of points at an exceptionally high metric value for any particular $S_g$ value (see Figs 12a, 12b, 12c, 12f, 12g and 12h). Instead, it seems to be an individual choice of these models per the $T_e$ field. For example, in the case of non-blurred images (evaluation using $J$), more strict models (Models 1 and 2) stick to specific $S_g$ values (see Fig 12a). For blurred images of the same strict models, the spectrum of well-performing $S_g$ values increases, but it still does not tend to one optimal value (see Fig 12b). The blurring of the images spatially diffuses the pixels, and the actual structure of the gas finger becomes less relevant, which makes up for the conceptual weakness of Models 1 and 2, allowing them to cope with more varied $S_g$ values. In other words, conceptually strong models are more flexible in their choice of $S_g$ values. This is further supported by the observed spread of $S_g$ values for Model 3 with $nb = 8$ (Fig 12c), which produced a gas finger with a close resemblance to the original experimental image for 10-A (see Fig 4 and 9).

In spite of the flexibility of choice of $S_g$ values, conceptually strong models are expected to favour a particular $S_g$ value. For Model 3, which ranks best in most scenarios of the table specified by Fig 6, the sub-version with $nb = 50$ does favour a single $S_g$ value (see Figs 12d, 12e, and 12i). However, this optimal $S_g$ value is not always realistic. For example, the converged $S_g$ value for Model 3 with $nb = 50$ is 0.12 for experiment 250-A (see Fig 12i). [55] reported typical $S_g$ values between 0.20 to 0.4 for the inner core and 0.03 to 0.20 for the outer shell of each gas finger, from the high injection rate

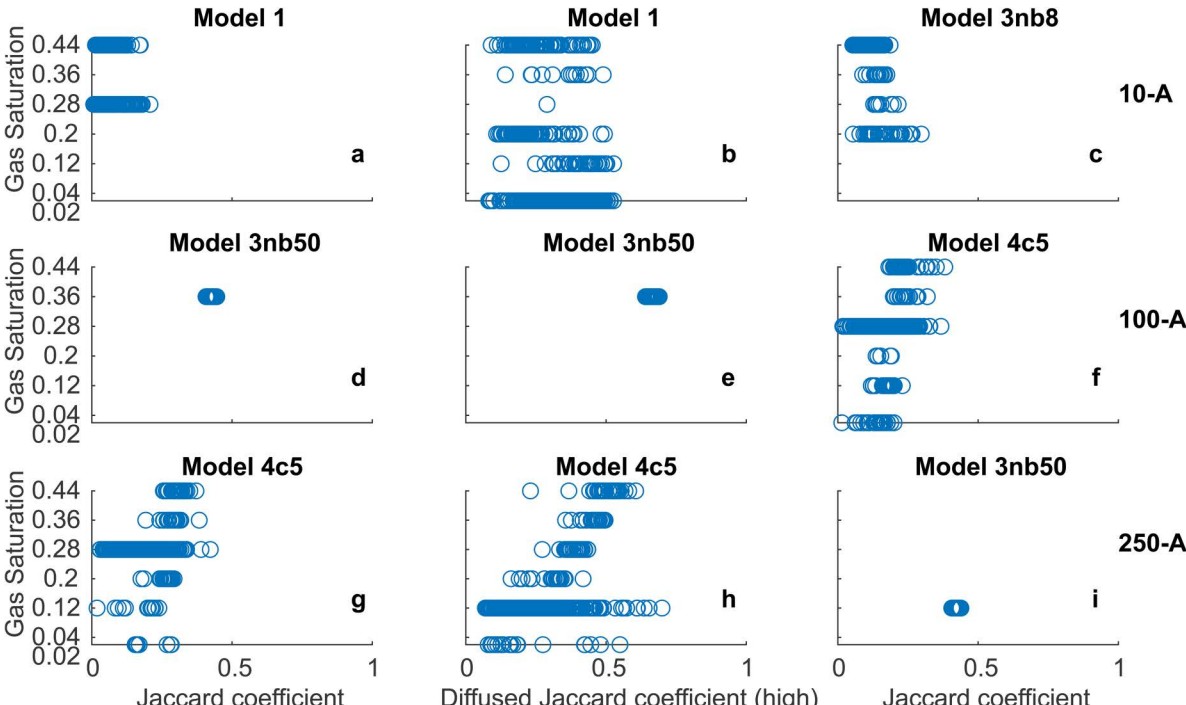

**Fig 12. A sample of nine plots showing the gas saturation distribution per model (sub-) version for all 500 realisations over the respective metric values for experiments 10-A, 100-A and 250-A.** The title of the subplots 3nb8 and 3nb50 stands for Model 3 with $nb = 8$ and $nb = 50$, respectively. The title of the subplots 4c5 stands for Model 4 with $c$ value 5.

(100 ml/min, 250 ml/min and 498 ml/min) experimental triplicate of [4]. Thus, the value of $S_g$ = 0.12 for the entire gas cluster is substantially lower than that observed and reported in [55], which implies that the model overestimates the overall extent of the gas distribution. As earlier discussed in Sect 4.2, Model 3 does not adequately predict the shape and structure of the gas clusters consisting of multiple fingers. Thus, the favoured $S_g$ value is merely the model's best attempt to fit the corresponding data.

For the close runner-up Model 4 with $c$ = 5, we do not observe any convergence to an optimal $S_g$ value (see Figs. 12f, 12g, and 12h). Recall that this model version's performance is highly sensitive to the input of the invasion threshold ($T_e$).

Therefore, the models apparently use the $S_g$ values to compensate either for their own conceptual weakness or for "poor" $T_e$ field inputs. Thus, from Fig 12, we can conclude that none of the models can predict the real physical $S_g$ values and thus are not recommended for $S_g$ calibration. As a possible way out, one could develop data assimilation or geostatistical inversion schemes for $T_e$ fields as already mentioned in Sect 4.3. Then, more plausible $S_g$ values could be obtained as only the conceptual weakness of models would remain as the major error source. Alternatively, model versions with variable gas-saturated blocks [71,47,72,73] are an optional extension of macroscopic IP models, which may be investigated for better calibration of $S_g$ values.

## 4.5 Summary of Findings

We summarise that Models 1 and 2 are unsuitable for use in transitional and continuous gas flow regimes, even with high levels of blurring in images (Sect 4.1). Models 3 and 4 perform better than Models 1 and 2 but do not accurately represent the gas finger patterns observed in the experiments (Sects 4.1 and 4.2). Model 3 is a good fit for experiments in the transitional gas flow regime (single slightly thick gas finger) but cannot appropriately predict the gas-finger patterns seen in the experiments of the continuous gas flow regime (multiple fingers) (Sect 4.2). Model 4 is a potential candidate for use in the transitional and continuous gas flow regimes, provided its rules are modified to reproduce the gas-flow behaviour at high injection rates (Sect 4.2). The modification of Model 4's underlying rules is beyond the scope of the present study. With blurring, i.e., at large scales where individual structures of the gas fingers are irrelevant, Models 3 and 4 may be used for continuous gas flow regimes (Sects 4.1 and 4.2). Their use would thus depend on the application. We also identify that the structure of the Te field is a critical input for a good performance of these models (Sect 4.3). The internal randomness of the invasion decision can partially compensate for the high uncertainty in the structure of the Te fields (Sects 4.1 and 4.2). Also, strategies like running multiple realisations of the Te field can help tackle this uncertainty of the Te fields. Further, we do not recommend these models for calibrating parameters like gas saturation (Sect 4.4), at least as long as there is a dominant uncertainty in $T_e$ fields.

## 5. Conclusions and outlook

In this study, we compared the performance of four macroscopic IP models against the data from nine experiments. The experiments featured gas injections in homogeneous water-saturated sand. For comparison, we used time-matching and (Diffused) Jaccard coefficient(s). For the first time, these models are tested in transitional and continuous gas-flow regimes. We identified the strengths and weaknesses of these modelling strategies for simulating gas flow in water-saturated sand. We also calibrated a few key parameters of these models.

Concerning the compared models, we conclude that Models 1 and 2 should not be used for the transitional and continuous regimes of gas flow discussed in this study. In particular, these models are completely weak for experiments at higher injection rates. In previous studies, IP models have been used extensively only in the capillary flow domain. Our results show that IP models at a macroscopic scale with variation as Model 3 can be used in the transitional gas flow regime but are unfit for use in the continuous gas flow regime. In their present state, Models 3 and 4 can be used with blurring for large-scale applications in the continuous gas flow regime, where the details of the gas-cluster structure are insignificant. Thus, the exact use depends on the specific application.

Models 3 and 4 perform better because they can partially account for the viscous effects found at high gas injection rates.

Although, Model 3 and Model 4 show some promise in performance, further research towards refining their rules for gas-invasion, water-re-invasion, finger branching, and so on needs to be done. A possible extension could be a mix of Model 3's rule of invading more blocks per step combined with a stochastic invasion rule similar to that of Model 4. The rule for this extension would also need to be adapted to better mimic the gas flow behaviour in the continuous flow regime, e.g., with finger invasion rules enabling the growth of multiple parallel thick fingers.

Regarding how to judge model-experiment agreement, we emphasize the importance of comparing results at an appropriate level of detail. In this context, blurring of images can be used as an efficient tool to reduce the detailed level of information in the images, depending on the application and the scale of interest. It is pointless to ask for a pixel-to-pixel match at and above the scale of the experiments used in this study, given the strong dependence of gas flow on pore-scale aspects of the porous medium (here: sand pack). This exercise can thus help justify the use of models like 3 or 4 for such applications.

A primary conclusion from our investigation of the sensitivity of the model parameters is that the underlying structure of the $T_e$ fields is a critical input for these models. Moreover, the best models (3 or 4) are also the most sensitive to this input. Further research could be conducted to identify the underlying structure of the $T_e$ fields, e.g., using geostatistical inversion methods.

Overall, our presented framework allowed us to compare several competing macroscopic IP models based on highly resolved image data. The employed Jaccard metric and its adjustable blurring parameter enabled an objective comparison that can focus on relevant scales of comparison and allows to fade out unnecessary pixel-wise detail. And finally, it provided relevant and valuable insights for further model development.

## Supporting information

**S1 Table. Gas Saturation Values Table containing the best-performing gas-saturation values per model version per experiment and for each metric used in this study.**
(PDF)

**S2 Fig. Experimental images and their blurred versions for experiments 10-B, 100-B and 250-B.**
(PDF)

**S3 Fig. Experimental images and their blurred versions for experiments 10-C, 100-C and 250-C.**
(PDF)

**S4 Fig. Best-fitting model realizations using the maximum Jaccard coefficient for experiments 10-B, 100-B and 250-B.**
(PDF)

**S5 Fig. Best-fitting model realizations using the maximum Jaccard coefficient for experiments 10-C, 100-C and 250-C.**
(PDF)

**S6 Fig. Best-fitting model realizations using the maximum Diffused Jaccard coefficient (low) for experiments 10-A, 100-A and 250-A.**
(PDF)

**S7 Fig. Best-fitting model realizations using the maximum Diffused Jaccard coefficient (low) for experiments 10-B, 100-B and 250-B.**
(PDF)

**S8 Fig. Best-fitting model realizations using the maximum Diffused Jaccard coefficient (low) for experiments 10-C, 100-C and 250-C.**
(PDF)

**S9 Fig. Best-fitting model realizations using the maximum Diffused Jaccard coefficient (med) for experiments 10-A, 100-A and 250-A.**
(PDF)

**S10 Fig. Best-fitting model realizations using the maximum Diffused Jaccard coefficient (med) for experiments 10-B, 100-B and 250-B.**
(PDF)

**S11 Fig. Best-fitting model realizations using the maximum Diffused Jaccard coefficient (med) for experiments 10-C, 100-C and 250-C.**
(PDF)

**S12 Fig. Best-fitting model realizations using the maximum Diffused Jaccard coefficient (high) for experiments 10-A, 100-A and 250-A.**
(PDF)

**S13 Fig. Best-fitting model realizations using the maximum Diffused Jaccard coefficient (high) for experiments 10-B, 100-B and 250-B.**
(PDF)

**S14 Fig. Best-fitting model realizations using the maximum Diffused Jaccard coefficient (high) for experiments 10-C, 100-C and 250-C.**
(PDF)

**S15 Appendix. Influence of inertial forces for the experiments of this study.**
(PDF)

## Acknowledgments

The authors sincerely thank the editor and reviewers for their constructive comments and valuable suggestions, which significantly improved the clarity and rigor of this manuscript. The authors also thank Dr Luciana Chavez Rodriguez from Wageningen University and Research, the Netherlands for her constructive feedback on an early version of this manuscript.

## Author contributions

**Conceptualization:** Ishani Banerjee, Anneli Guthke, Cole J.C. Van De Ven, Kevin G. Mumford, Wolfgang Nowak.

**Data curation:** Cole J.C. Van De Ven.

**Formal analysis:** Ishani Banerjee, Anneli Guthke.

**Funding acquisition:** Wolfgang Nowak.

**Investigation:** Ishani Banerjee.

**Methodology:** Ishani Banerjee, Anneli Guthke.

**Project administration:** Kevin G. Mumford, Wolfgang Nowak.

**Resources:** Wolfgang Nowak.

**Supervision:** Anneli Guthke, Kevin G. Mumford, Wolfgang Nowak.

**Visualization:** Ishani Banerjee.

**Writing – original draft:** Ishani Banerjee.

**Writing – review & editing:** Ishani Banerjee, Anneli Guthke, Cole J.C. Van De Ven, Kevin G. Mumford, Wolfgang Nowak.

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
