## [Decision Letter · Decision Letter 0]

3 Sep 2025

Dear Dr. Banerjee,

Thank you for submitting your manuscript to PLOS ONE. After careful consideration, we feel that it has merit but does not fully meet PLOS ONE’s publication criteria as it currently stands. Therefore, we invite you to submit a revised version of the manuscript.

**Between them the reviewers have raised detailed and thoughtful questions, comment and suggestions. In your revision, please address each in detail.**

We look forward to receiving your revised manuscript.

Kind regards,

Reuben O'Dea

Academic Editor

PLOS ONE

**Journal Requirements:**

1. When submitting your revision, we need you to address these additional requirements. Please ensure that your manuscript meets PLOS ONE's style requirements, including those for file naming. The PLOS ONE style templates can be found at https://journals.plos.org/plosone/s/file?id=wjVg/PLOSOne_formatting_sample_main_body.pdf and https://journals.plos.org/plosone/s/file?id=ba62/PLOSOne_formatting_sample_title_authors_affiliations.pdf 2. Thank you for stating the following financial disclosure: German Research Foundation (DFG) for financial support of this project within the Research Training Group GRK1829 “Integrated  Hydrosystem  Modelling” and the Cluster of Excellence EXC 2075 “Data-integrated Simulation Science (SimTech)” at the University of Stuttgart under Germany’s Excellence Strategy - EXC 2075 - 39074001   Please state what role the funders took in the study.  If the funders had no role, please state: "The funders had no role in study design, data collection and analysis, decision to publish, or preparation of the manuscript." If this statement is not correct you must amend it as needed. Please include this amended Role of Funder statement in your cover letter; we will change the online submission form on your behalf. 3. Thank you for stating the following in the Acknowledgments Section of your manuscript: The authors would like to thank the German Research Foundation (DFG) for financial 851 support of this project within the Research Training Group GRK1829 “Integrated 852 Hydrosystem Modelling” and the Cluster of Excellence EXC 2075 “Data-integrated 853 Simulation Science (SimTech)” at the University of Stuttgart under Germany’s 854 Excellence Strategy - EXC 2075 - 39074001. The authors would also like to thank Dr 855 Luciana Chavez Rodriguez from the Wageningen University and Research, 856 Netherlands for her constructive feedback on an early version of this manuscript. We note that you have provided funding information that is not currently declared in your Funding Statement. However, funding information should not appear in the Acknowledgments section or other areas of your manuscript. We will only publish funding information present in the Funding Statement section of the online submission form. Please remove any funding-related text from the manuscript and let us know how you would like to update your Funding Statement. Currently, your Funding Statement reads as follows: German Research Foundation (DFG) for financial support of this project within the Research Training Group GRK1829 “Integrated  Hydrosystem  Modelling” and the Cluster of Excellence EXC 2075 “Data-integrated Simulation Science (SimTech)” at the University of Stuttgart under Germany’s Excellence Strategy - EXC 2075 - 39074001 Please include your amended statements within your cover letter; we will change the online submission form on your behalf. 4. Thank you for uploading your study's underlying data set. Unfortunately, the repository you have noted in your Data Availability statement does not qualify as an acceptable data repository according to PLOS's standards. At this time, please upload the minimal data set necessary to replicate your study's findings to a stable, public repository (such as figshare or Dryad) and provide us with the relevant URLs, DOIs, or accession numbers that may be used to access these data. For a list of recommended repositories and additional information on PLOS standards for data deposition, please see https://journals.plos.org/plosone/s/recommended-repositories. 5. When completing the data availability statement of the submission form, you indicated that you will make your data available on acceptance. We strongly recommend all authors decide on a data sharing plan before acceptance, as the process can be lengthy and hold up publication timelines. Please note that, though access restrictions are acceptable now, your entire data will need to be made freely accessible if your manuscript is accepted for publication. This policy applies to all data except where public deposition would breach compliance with the protocol approved by your research ethics board. If you are unable to adhere to our open data policy, please kindly revise your statement to explain your reasoning and we will seek the editor's input on an exemption. Please be assured that, once you have provided your new statement, the assessment of your exemption will not hold up the peer review process. 6. Please amend either the abstract on the online submission form (via Edit Submission) or the abstract in the manuscript so that they are identical. 7. Please upload a copy of S14 Figure, to which you refer in your text on page 22. If the figure is no longer to be included as part of the submission please remove all reference to it within the text. 8. If the reviewer comments include a recommendation to cite specific previously published works, please review and evaluate these publications to determine whether they are relevant and should be cited. There is no requirement to cite these works unless the editor has indicated otherwise. 

Reviewers' comments:

**Comments to the Author**

1. Is the manuscript technically sound, and do the data support the conclusions?

Reviewer #1: Yes

Reviewer #2: Yes

2. Has the statistical analysis been performed appropriately and rigorously?

Reviewer #1: Yes

Reviewer #2: Yes

3. Have the authors made all data underlying the findings in their manuscript fully available?

Reviewer #1: Yes

Reviewer #2: Yes

4. Is the manuscript presented in an intelligible fashion and written in standard English?

Reviewer #1: Yes

Reviewer #2: Yes

**Reviewer #1:** This study compares four macroscopic Invasion Percolation (IP) models for simulating gas flow in water-saturated porous media beyond the discontinuous regime. Using light transmission imaging and randomised entry pressure fields, the models are tested against experiments in transitional and continuous flow regimes. A diffused Jaccard coefficient is used to evaluate performance. Results show that some IP models are effective outside their traditional application range, and that initial entry pressure fields have a significant influence on outcomes. The study presents a generalizable framework for model evaluation using high-resolution spatial and temporal data. A minor revision is recommended to improve clarity and quality.

1. Introduction:

In Lines 74-75, as for the discontinuous flow regime, discrete gas bubbles could be modelled by coupling buoyancy-driven bubbly flow in porous medium with Navier-Stokes equations rather than capillary-viscous-inertia-driven air slugs and air continuous flow. This model has been well-developed in COMOSAL.

Recommended work: Ma, Y., Yan, G., & Scheuermann, A. (2022). Discrete bubble flow in granular porous media via multiphase computational fluid dynamic simulation. Frontiers in Earth Science, 10, 947625.

2. Methodology:

Figure 1 is a good illustration of IP, while you forgot to provide a photo of your IP model in both the main text and the appendix. It would be beneficial to draw the dimensions of the IP model, the initial and boundary conditions applied to it, and include a scale bar on the IP images.

Additionally, the optical calibration between physical length and pixel should be provided, along with the imaging methods (camera, resolution, frame per second, etc.), image processing (grayscale threshold for binarization, noise filter algorithm, etc.), and image analysis. Not all details are necessary, but some key information should be mentioned for experimental replication.

Line 452: No indentation ahead of notation starting with “where”. Please fixe the same issues elsewhere.

3. Results and discussions:

Have you paid attention to the high gas injection-induced inertia effect in two immiscible phase seepage flow in porous media, as high injection flow rate may trigger dynamic nonequilibrium capillary water flow. These could be verified by comparing the Reynolds number and the Weber number to assess the relative contributions of inertial, viscous, and capillary forces.

Recommended publication: Yan, G., Li, Z., Bore, T., Torres, S. A. G., Scheuermann, A., & Li, L. (2022). A lattice Boltzmann exploration of two-phase displacement in 2D porous media under various pressure boundary conditions. Journal of Rock Mechanics and Geotechnical Engineering, 14(6), 1782-1798.

4. Conclusions:

Please restructure the conclusion as follows:

Paragraph 1, Overall Summary: Begins with a broad overview of the topic, outlining the scope, key themes, and the progression of the review. Sets the context by connecting existing knowledge with emerging challenges or gaps.

Paragraph 2, Thematic or Point-by-Point Breakdown: Followed by a structured list of conclusions or insights, usually numbered or bulleted. Each point presents a distinct observation, finding, or implication, often supported by comparisons or qualifications. The list moves logically from theoretical foundations to experimental methods and finally to practical or applied considerations.

Paragraph 3, Final Integrative Conclusion: Ends with a synthesis or reflection, often forward-looking. Highlights the practical relevance, implications for future work, or broader significance of the findings.

**Reviewer #2:** This manuscript presents a comparative framework for evaluating four macroscopic invasion percolation (IP) models against experimental data from gas injection in water-saturated sand across transitional and continuous flow regimes. The authors use a diffused Jaccard coefficient approach to rank model performance and identify that Models 3 and 4 (which incorporate rules for multiple block invasion and stochastic selection, respectively) outperform traditional IP models in these previously unexplored flow regimes.

- Novel Evaluation Framework: The introduction of the diffused Jaccard coefficient and a robust, image-based comparison method provides a significant advancement for model-data comparison in porous media research.

- Clear Problem Definition: The manuscript addresses a well-defined gap in the literature regarding the suitability of IP models for flow regimes beyond traditional discontinuous cases.

- Comprehensive Experimental Basis: Nine well-characterized experimental datasets, covering triplicates at different flow rates, offer a strong basis for benchmarking.

Major comments:

- The abstract is overly technical and method-focused. It under-emphasizes the broader significance of the work.

Revise to highlight (a) the problem addressed, (b) the novelty of the framework, (c) the key results (e.g., Model 3 generally outperforms others), and (d) implications for practical applications such as CO₂ storage or groundwater protection.

- Limited Physical Understanding: The manuscript lacks sufficient mechanistic insight into why certain models perform better. The authors acknowledge that Models 3 and 4 can "partially consider viscous effects" but do not provide clear physical justification for how multiple block invasion or stochastic selection actually represents viscous processes.

-The manuscript claims that the framework is generalizable beyond gas-water systems, but this is not strongly substantiated. Provide concrete examples of how this framework could be applied to other multiphase systems (e.g., oil-water, CO₂ sequestration) or even to data-driven models.

- Address Model Limitations: Explicitly discuss the physical meaning of unrealistic gas saturation values and their implications for model applicability.

- Expand Experimental Scope: Include additional porous media types or acknowledge this as a significant limitation.

Minor comments:

- The manuscript is generally well-written, though some sections (especially methodological details) could be condensed or clarified for broader accessibility.

- Extend analysis to 3D models or justify 2D limitation

- Systematic evaluation of parameter sensitivity

- Consider additional comparison metrics beyond Jaccard coefficient

**Do you want your identity to be public for this peer review?** For information about this choice, including consent withdrawal, please see our Privacy Policy

Reviewer #1: No

Reviewer #2: No

---

## [Author Response · Author response to Decision Letter 1]

7 Dec 2025

The response to the editor and the Reviewers comments has been uploaded as the document : Response Letter_final.docx

Please find the same here:

Response to Editor and Reviewers

Dear Prof. Reuben O'Dea,

We would like to sincerely thank you and the reviewers for your thoughtful and constructive comments on our manuscript : PONE-D-25-31748: “A framework for objectively comparing competing invasion percolation models based on highly-resolved image data”. We have carefully considered all the feedback and revised the manuscript accordingly. Below, we provide detailed responses to each comment, along with a summary of the changes made in the revised version.

We believe these revisions have significantly improved the quality and clarity of our manuscript.

General Comments

We have made the following general improvements to the manuscript based on the reviewers’ feedback:

Journal Requirements:

Thank you for the comment. This has been duly noted.

German Research Foundation (DFG) for financial support of this project within the Research Training Group GRK1829 “Integrated Hydrosystem Modelling” and the Cluster of Excellence EXC 2075 “Data-integrated Simulation Science (SimTech)” at the University of Stuttgart under Germany’s Excellence Strategy - EXC 2075 - 39074001

The authors would like to thank the German Research Foundation (DFG) for financial 851 support of this project within the Research Training Group GRK1829 “Integrated 852 Hydrosystem Modelling” and the Cluster of Excellence EXC 2075 “Data-integrated 853 Simulation Science (SimTech)” at the University of Stuttgart under Germany’s 854 Excellence Strategy - EXC 2075 - 39074001. The authors would also like to thank Dr 855 Luciana Chavez Rodriguez from the Wageningen University and Research, 856 Netherlands for her constructive feedback on an early version of this manuscript.

German Research Foundation (DFG) for financial support of this project within the Research Training Group GRK1829 “Integrated Hydrosystem Modelling” and the Cluster of Excellence EXC 2075 “Data-integrated Simulation Science (SimTech)” at the University of Stuttgart under Germany’s Excellence Strategy - EXC 2075 - 39074001

Ans:

Thank you for the information. The acknowledgement section has been modified. Our funding statement should be as follows:

“The authors would like to thank the German Research Foundation (DFG) for financial support of this project within the Research Training Group GRK1829 “Integrated Hydrosystem Modelling” and the Cluster of Excellence EXC 2075 “Data-integrated Simulation Science (SimTech)” at the University of Stuttgart under Germany’s Excellence Strategy - EXC 2075 - 39074001.”

4. Thank you for uploading your study's underlying data set. Unfortunately, the repository you have noted in your Data Availability statement does not qualify as an acceptable data repository according to PLOS's standards.

Ans: As per the correspondence with the editor, the repositories would be made public upon acceptance.

Ans:

The concern has been duly noted. The repository will be made public as soon as the paper gets accepted. The background processes for the same have been dealt with.

6. Please amend either the abstract on the online submission form (via Edit Submission) or the abstract in the manuscript so that they are identical.

Ans:

We have ensured that the Abstracts are identical.

7. Please upload a copy of S14 Figure, to which you refer in your text on page 22. If the figure is no longer to be included as part of the submission, please remove all reference to it within the text.

Ans:

Thank you for observing the missing figure in the Supporting information. The S14 Figure is now uploaded.

Response to Reviewer #1

Reviewer #1: This study compares four macroscopic Invasion Percolation (IP) models for simulating gas flow in water-saturated porous media beyond the discontinuous regime. Using light transmission imaging and randomised entry pressure fields, the models are tested against experiments in transitional and continuous flow regimes. A diffused Jaccard coefficient is used to evaluate performance. Results show that some IP models are effective outside their traditional application range, and that initial entry pressure fields have a significant influence on outcomes. The study presents a generalizable framework for model evaluation using high-resolution spatial and temporal data. A minor revision is recommended to improve clarity and quality.

We would like to thank you for your time and feedback on our manuscript. Please find the answers to the individual questions raised below.

1. Introduction:

In Lines 74-75, as for the discontinuous flow regime, discrete gas bubbles could be modelled by coupling buoyancy-driven bubbly flow in porous medium with Navier-Stokes equations rather than capillary-viscous-inertia-driven air slugs and air continuous flow. This model has been well-developed in COMOSAL.

Recommended work: Ma, Y., Yan, G., & Scheuermann, A. (2022). Discrete bubble flow in granular porous media via multiphase computational fluid dynamic simulation. Frontiers in Earth Science, 10, 947625.

Ans:

We thank the reviewer for suggesting this reference. We have read the recommended paper carefully and found that its modelling approach is indeed relevant to our study. Accordingly, we have incorporated this technique into the revised manuscript in lines 101-103.

“Modified continuum models formulated at the pore scale—using the Navier–Stokes equations to represent bubbly gas transport—have also been used to simulate discontinuous gas flow; however, this approach also has high computational cost, thus limiting their practical applicability at larger scales [75].”

2. Methodology:

Figure 1 is a good illustration of IP, while you forgot to provide a photo of your IP model in both the main text and the appendix. It would be beneficial to draw the dimensions of the IP model, the initial and boundary conditions applied to it, and include a scale bar on the IP images.

Additionally, the optical calibration between physical length and pixel should be provided, along with the imaging methods (camera, resolution, frame per second, etc.), image processing (grayscale threshold for binarization, noise filter algorithm, etc.), and image analysis. Not all details are necessary, but some key information should be mentioned for experimental replication.

Line 452: No indentation ahead of notation starting with “where”. Please fixe the same issues elsewhere.

Ans:

We thank the reviewer for raising this point. The intention of the remark is not entirely clear to us, so we address it under two possible interpretations below.

(A) If the reviewer is referring to the computational models (Invasion Percolation models) used in the study, our response is as follows:

We would like to clarify that the Invasion-Percolation (IP) models employed in our model comparison framework in this study have been well established and thoroughly described in the cited literature. Also, we have included all the relevant information on the general background of these models in Lines 124- 181. For the 4 specific models we have further included the description with equations in Sections 2.2-2.5. The model image outputs in the figure, are from these four corresponding models. All input parameter values used in the models are provided in Table 1.

The domain size of both the experiments and the models was kept consistent at 250 mm × 250 mm, corresponding to the size of the output images from the models. These dimensions are maintained in the figures of the manuscript; hence, a scale bar has not been included.

Because the primary aim of this work is to develop and demonstrate the evaluation framework—rather than to elaborate on the experimental setup or modelling procedures—we refer readers to the original publication for full technical details needed for replication.

(B) Alternatively, if the reviewer is referring to the experiments, then our response is as follows:

We have added the following lines to the revised manuscript in line numbers 282-294:

“In this method, the back of the cell is lit, and digital images or videos of gas injection are recorded. In the experiments used in this study, imaging was performed using a high-resolution camera (Canon EOS 6D equipped with a Canon EF 17–40 mm lens). The videos were captured at a resolution of 1920 × 1080 pixels and a frame rate of 29.97 frames s−1 for the duration of each experiment. The recordings were subsequently processed to extract still-frame images, which were cropped to include only the face of the cell and then converted to grayscale intensity fields. Individual pixel intensity values of these raw images are averaged over a block size of 1 × 1 mm, and the intensity values of the block are used to calculate the optical density (OD) [42] values. A median filter is applied to a 3 × 3 pixel (0.75 × 0.75 mm) neighbourhood so that the resulting OD field reflects variations over areas comparable to, but not smaller than, the diameter of an individual grain.”

The experimental data processing methodology is described in detail by Van de Ven et al. (2019). In this manuscript, we provide only the information necessary to support the evaluation framework presented here, but now adding a few clarifying details as recommended by the reviewer.

Because the primary aim of this work is to develop and demonstrate the evaluation framework—rather than to elaborate on the experimental setup or modelling procedures—we refer readers to the original publication for full technical details needed for replication.

The indentation issues noted by the reviewer have been carefully corrected in the revised manuscript.

3. Results and discussions:

Have you paid attention to the high gas injection-induced inertia effect in two immiscible phase seepage flow in porous media, as high injection flow rate may trigger dynamic nonequilibrium capillary water flow. These could be verified by comparing the Reynolds number and the Weber number to assess the relative contributions of inertial, viscous, and capillary forces.

Recommended publication: Yan, G., Li, Z., Bore, T., Torres, S. A. G., Scheuermann, A., & Li, L. (2022). A lattice Boltzmann exploration of two-phase displacement in 2D porous media under various pressure boundary conditions. Journal of Rock Mechanics and Geotechnical Engineering, 14(6), 1782-1798.

Ans:

We thank the reviewer for this comment. In determining the Reynolds and Weber numbers for our experiments, we drew on the “near-source” and “far-source” conceptual framework for gas injection into water-saturated porous media introduced by Selker et al. (2006). This framework guided our use of the inflating-sphere approximation to estimate the radii at which inertial, viscous, and capillary forces become dominant. We have added a brief explanation of this approach in S15 Appendix of the revised manuscript.

“Here, we examine how inertial forces influence the experimental conditions in our study. We conceptualize the injected gas as forming an initial spherical interface with a radius set by the injection needle. Our aim is to determine the radius of this expanding gas sphere beyond which inertial effects become small compared with viscous and capillary forces. To do this, we evaluate the Reynolds number (Re, ratio of inertial to viscous forces) and the Weber number (We, ratio of inertial to capillary forces):

Re = (ρ_g Q r) / (A μ_g)

We = (ρ_g Q^2 r) / (A^2 σ)

where ρ_g and μ_g are the density and dynamic viscosity of gas (air), Q is the gas-injection rate, r is the radius of the gas sphere, A is the interfacial area of the sphere, and σ is the air–water interfacial tension.

We rewrite the equations for the radius r of the expanding gas sphere for the cases Re = 1 and We = 1:

r(Re = 1) = (ρ_g Q) / (4 π μ_g)

r(We = 1) = ( (ρ_g Q^2) / (16 π^2 σ) )^(1/3)

Using the values ρ_g = 3 kg/m³, μ_g = 0.019 × 10⁻³ N·s/m², and σ = 0.071 N/m, we obtain the following results for the different experiments:

Injection rate (ml/min) r (Re = 1) [mm] r (We = 1) [mm]

10 2.09 0.02

100 20.9 0.07

250 52.4 0.13

We observe that even for the highest injection rate (250 ml/min), inertial forces exceed capillary forces only within a region smaller than 0.2 mm from the gas-injection point, which is less than the mean grain diameter of the sand (d₅₀ = 0.713 ± 0.023 mm). This indicates that, at both pore and continuum scales, surface tension dominates over gas inertia. At the same injection rate, inertial forces exceed viscous forces up to roughly 50 mm from the inlet, but this range is still well below the size of the sandbox (125 mm to each side), and capillary forces remain the controlling factor throughout. Overall, these estimates show that inertial forces do not determine the displacement patterns observed in our experiments; capillary and viscous forces do.”

4. Conclusions:

Please restructure the conclusion as follows:

Paragraph 1, Overall Summary: Begins with a broad overview of the topic, outlining the scope, key themes, and the progression of the review. Se

---

## [Decision Letter · Decision Letter 1]

22 Dec 2025

A framework for objectively comparing competing invasion percolation models based on highly-resolved image data

PLOS One

Dear Dr. Banerjee,

Thank you for submitting your manuscript to PLOS ONE and for your careful and thorough response to the reviewers. Nevertheless, there remains one additional point raised by reviewer 1 that you should consider addressing. Therefore, we invite you to submit a revised version of the manuscript that addresses this point.

We look forward to receiving your revised manuscript.

Kind regards,

Reuben O'Dea

Academic Editor

PLOS One

Journal Requirements:

Reviewers' comments:

Reviewer's Responses to Questions

**Comments to the Author**

Reviewer #1: (No Response)

Reviewer #2: All comments have been addressed

2. Is the manuscript technically sound, and do the data support the conclusions?

Reviewer #1: Yes

Reviewer #2: Yes

3. Has the statistical analysis been performed appropriately and rigorously?

Reviewer #1: Yes

Reviewer #2: Yes

4. Have the authors made all data underlying the findings in their manuscript fully available?

Reviewer #1: Yes

Reviewer #2: Yes

5. Is the manuscript presented in an intelligible fashion and written in standard English?

Reviewer #1: Yes

Reviewer #2: Yes

Reviewer #1: The authors have mainly addressed the comments from the last round of review. I have no additional comments on it except for one raised by the other reviewer (Reviewer #2).

The authors answered that heterogeneity in the initial entry-pressure field strongly influences model performance (Lines 47-48). You answered this well in your response letter and included additional information in the abstract. However, you forgot to mention this part in the main text and only put minor information in Line 742 (the uncertainty of the geological heterogeneity in their experimental setup). Could you please add the essential findings and discussion not only in the abstract and response letter but also in the main text, in the results and discussion chapters?

Also, another good study investigated the role of local heterogeneity in the initial entry pressure and scale effects in the capillarity-saturation relationship, which may help to support this finding (“the initial entry-pressure field strongly influences model performance”).

Yan, G., Bore, T., Schlaeger, S., Scheuermann, A., & Li, L. (2022). Investigating scale effects in soil water retention curve via spatial time domain reflectometry. Journal of Hydrology, 612, 128238.

Reviewer #2: (No Response)

**Do you want your identity to be public for this peer review?** For information about this choice, including consent withdrawal, please see our Privacy Policy

Reviewer #1: No

Reviewer #2: **Yes:** Sukirt Thakur

---

## [Author Response · Author response to Decision Letter 2]

5 Feb 2026

Dear Prof. Reuben O'Dea

We would like to sincerely thank you and the reviewer 1 for your constructive comment on our manuscript: PONE-D-25-31748: “A framework for objectively comparing competing invasion percolation models based on highly-resolved image data”. We have carefully revised the manuscript accordingly. Below, we provide a detailed response to the comment, along with a summary of the changes made in the revised version.

Response to Reviewer 1:

Reviewer #1: The authors have mainly addressed the comments from the last round of review. I have no additional comments on it except for one raised by the other reviewer (Reviewer #2).

The authors answered that heterogeneity in the initial entry-pressure field strongly influences model performance (Lines 47-48). You answered this well in your response letter and included additional information in the abstract. However, you forgot to mention this part in the main text and only put minor information in Line 742 (the uncertainty of the geological heterogeneity in their experimental setup). Could you please add the essential findings and discussion not only in the abstract and response letter but also in the main text, in the results and discussion chapters?

Also, another good study investigated the role of local heterogeneity in the initial entry pressure and scale effects in the capillarity-saturation relationship, which may help to support this finding (“the initial entry-pressure field strongly influences model performance”).

Yan, G., Bore, T., Schlaeger, S., Scheuermann, A., & Li, L. (2022). Investigating scale effects in soil water retention curve via spatial time domain reflectometry. Journal of Hydrology, 612, 128238.

Ans: Thank you for your question, which prompted us to re-examine the terminology used in the manuscript. In doing so, we realized that our parallel use of entry pressure fields and invasion threshold fields could lead to confusion, even though this was not explicitly raised.

To clarify this point, we now consistently use the term invasion threshold fields throughout the abstract and main text. At the scale of our model, the invasion threshold represents an effective quantity that combines the local entry pressure field with the hydrostatic pressure field.

Further, the detailed analyses describing how model performance depends on the invasion threshold fields were already presented in Sections 4.1.1 and 4.1.2. To improve readability and better guide the reader, we have now explicitly synthesized these existing results and added a short summary in Section 4.2, highlighting the central role of invasion threshold uncertainty in model performance.

“The analysis across 500 $T_e$ realisations in Section 4.1.2 demonstrates that model performance is highly conditional on the invasion threshold field. Deterministic model variants perform exceptionally well only for a narrow subset of favourable $T_e$ fields, while more flexible formulations show greater robustness when the $T_e$ field poorly represents the experimental medium. This sensitivity confirms that $T_e$ is a critical input for the models.”

We thank the reviewer for the helpful suggestion of the study: “Yan, G., Bore, T., Schlaeger, S., Scheuermann, A., & Li, L. (2022). Investigating scale effects in soil water retention curve via spatial time domain reflectometry. Journal of Hydrology, 612, 128238”. However, while experimental measurements may indeed be subject to such uncertainties, our discussion is framed from the perspective of model-concept uncertainty, focussing on how model performance is affected when the true invasion threshold field cannot be represented, rather than on experimental measurement approaches, and we therefore do not cite the study.

We hope that the revised manuscript meets your expectations and look forward to your favorable consideration.

Sincerely,

Dr.-Ing. Ishani Banerjee

---

## [Editor Report · Decision Letter 2]

11 Feb 2026

A framework for objectively comparing competing invasion percolation models based on highly-resolved image data

PONE-D-25-31748R2

Dear Dr. Banerjee,

We’re pleased to inform you that your manuscript has been judged scientifically suitable for publication and will be formally accepted for publication once it meets all outstanding technical requirements.

Kind regards,

Reuben O'Dea

Academic Editor

PLOS One
---

## [Editor Report · Acceptance letter]

PONE-D-25-31748R2

PLOS One

Dear Dr. Banerjee,

I'm pleased to inform you that your manuscript has been deemed suitable for publication in PLOS One. Congratulations! Your manuscript is now being handed over to our production team.

Kind regards,

on behalf of

Dr. Reuben O'Dea

Academic Editor

PLOS One